# ADVERSARIAL RADEMACHER COMPLEXITY OF DEEP NEURAL NETWORKS

## ABSTRACT

Deep neural networks are vulnerable to adversarial attacks. Adversarial training is one of the most effective algorithms to increase the model's robustness. However, the trained models cannot generalize well to the adversarial examples on the test set. In this paper, we study the generalization of adversarial training through the lens of adversarial Rademacher complexity. Current analysis of adversarial Rademacher complexity is up to two-layer neural networks. In adversarial settings, one major difficulty of generalizing these results to deep neural networks is that we cannot peel off the layer as the classical analysis for standard training. We provide a method to overcome this issue and provide upper bounds of adversarial Rademacher complexity of deep neural networks. Similar to the existing bounds of standard Rademacher complexity of neural nets, our bound also includes the product of weight norms. We provide experiments to show that the adversarially trained weight norms are larger than the standard trained weight norms, thus providing an explanation for the bad generalization performance of adversarial training.

## 1 INTRODUCTION

Deep neural networks (DNNs) (Krizhevsky et al. (2012); Hochreiter & Schmidhuber (1997)) have become successful in many machine learning tasks such as computer vision (CV) and natural language processing (NLP). But they are shown to be vulnerable to adversarial examples (Szegedy et al. (2013); Goodfellow et al. (2014)). A well-trained model can be easily attacked by adding a small perturbation to the original data. Adversarial training is one of the most effective algorithms to defend against adversarial attacks. However, generalization is one of the main issues of adversarial training. An adversarially-trained model will overfit the adversarial examples on the training dataset, and it cannot generalize well to the adversarial examples on the testset. For example, in the experiment of training ResNet (He et al. (2016)) on CIFAR-10 (Krizhevsky et al. (2009)), Projected gradient descent (PGD) adversarial training achieves 100% robust accuracy on the training set, but it only gets 45% robust accuracy on the test set (Madry et al. (2017)). Recent works (Gowal et al. (2020); Rebuffi et al. (2021)) mitigate the overfitting issue, but it still has a 20% generalization gap between robust test accuracy (60%) and training accuracy (80%). On the other hand, a standard trained model can generalize well to the test set with a 5% generalization gap. To understand why the generalization of adversarial training behaves differently from standard training, we study the generalization issue of adversarial training through the lens of Rademacher complexity.

In classical machine learning theory, Rademacher complexity measures the generalization capacity of machine learning models. For depth-$d$ neural networks, assuming that the weight matrices $W_1, W_2, \cdots, W_d$ in each of the $d$ layers have Frobenius norms bounded by $M_1, \cdots, M_d$, and all the data $\boldsymbol{x}$ have $\ell_2$-norm bounded by $B$, given $m$ training samples, the generalization gap between population risk and empirical risk scales as $\mathcal{O}(B2^d \prod_{l=1}^d M_l/\sqrt{m})$ with high probability (Neyshabur et al. (2015)). Another works provide different norm-based complexity, such as $\| \cdot \|_{1,\infty}$-norm (Bartlett & Mendelson (2002)) and spectral norm (Bartlett et al. (2017)). The work of (Golowich et al. (2018)) reduces the dependence on depth-$d$ from $2^d$ to $\sqrt{d}$. The proofs of the above bounds are based on the induction on layers, which is also called the 'peeling off' techniques. For more details, see section 3.

In adversarial training, adversarial Rademacher complexity was first introduced in (Yin et al. (2019); Khim & Loh (2018)) to measure the robust generalization gap. They prove that the robust generalization gap of linear function $(\boldsymbol{x} \to w^T \boldsymbol{x})$ scales as $\mathcal{O}((B+\epsilon)M/\sqrt{m})$, where $M$ is the upper bound of norm of the weights $w$ and $\epsilon$ is the perturbation intensity of adversarial attacks. Awasthi et al. (2020) provides an upper bound in two-layers neural network cases. For depth-2, width-$h$ neural networks, with high probability, the generalization gap scales as $\mathcal{O}((B+\epsilon)\sqrt{hq}M_1M_2\sqrt{\log m/m})$, where $q$ is the dimension of the data $\boldsymbol{x}$.

One might think it is straightforward to use the induction methods in (Neyshabur et al. (2015); Golowich et al. (2018)) to extend the adversarial Rademacher complexity in (Yin et al. (2019); Khim & Loh (2018)) to multi-layers cases. However, it seems challenging to apply their induction methods to adversarial cases. Let the adversarial loss be $\max_{\|x-x'\|\leq\epsilon} \ell(f(\boldsymbol{x}'), y)$ and $f$ is a DNN. We cannot peel off the layer because of the max operation in the adversarial loss. The work of (Khim & Loh (2018)) and (Gao & Wang (2021)) also indicate the difficulty of analyzing adversarial Rademacher complexity of DNNs. They analyze other variants of adversarial Rademacher complexity, which are quite different from the original adversarial Rademacher complexity. See more detailed discussions of these works in section 3. To our knowledge, direct analysis of the original adversarial Rademacher complexity is largely missing.

In this paper, we analyze the adversarial Rademacher complexity of deep neural networks. Specifically, for depth-$d$, width-$h$ fully connected neural networks, with high probability, the robust generalization gap scales as

$$\mathcal{O}\left(\frac{(B+\epsilon)h\sqrt{d\log d}\prod_{l=1}^{d}M_l}{\sqrt{m}}\right).$$

Similar to the existing bounds of standard Rademacher complexity of deep neural nets, the bound includes the product of weight norms $\prod_{l=1}^{d}\|W_l\|$, but they are trained by different algorithms. We empirically show that the adversarially trained weight norms are larger than the standard trained weight norms, which provide an explanation why adversarial training did not generalize well. Our contributions are listed as follow:

1. We provide a method and give upper and lower bounds for the adversarial Rademacher complexity of deep neural nets. Compared to standard Rademacher complexity, the bound has a higher-order dependence on the depth and width and an additional factor $\epsilon$.

2. We provide experiments to analyze the relationship between the generalization gap and the adversarial Rademacher complexity. We show that one of the reasons why adversarial training cannot generalize well is the large weight norms of an adversarially-trained model.

## 2 PRELIMINARIES

### 2.1 GENERALIZATION GAP AND RADEMACHER COMPLEXITY

**Generalization Gap.** We start from the classical machine learning framework. Let $\mathcal{F}$ be the hypothesis class (e.g. Linear functions, Neural networks). The goal of the learning problem is to find $f \in \mathcal{F}$ to minimize the population risk $R(f) = \mathbb{E}_{(\boldsymbol{x},y)\sim\mathcal{D}}[\ell(f(\boldsymbol{x}),y)]$, where $\mathcal{D}$ is the true distribution, $\ell(\cdot)$ is the loss function. Since $\mathcal{D}$ is unknown, we minimize the empirical risk in practice. Given $m$ i.i.d samples $S = \{(\boldsymbol{x}_1, y_1), \cdots, (\boldsymbol{x}_m, y_m)\}$, the empirical risk is $R_m(f) = \frac{1}{m}\sum_{i=1}^{m}[\ell(f(\boldsymbol{x}_i), y_i)]$. The generalization gap or generalization error is defined as follow:

$$\text{Generalization Gap} := R(f) - R_m(f).$$

**Rademacher Complexity.** A classical measure of the generalization error is Rademacher complexity (Bartlett & Mendelson (2002)). Given the hypothesis class $\mathcal{H}$, the (empirical) Rademacher complexity is defined as

$$\mathcal{R}_\mathcal{S}(\mathcal{H}) = \mathbb{E}_\sigma \frac{1}{m}\Big[\sup_{h\in\mathcal{H}}\sum_{i=1}^{m}\sigma_i h(\boldsymbol{x}_i, y_i)\Big],$$

where $\sigma_i$ are i.i.d Rademacher random variables, i.e. $\sigma_i$ equals to $1$ or $-1$ with equal probability. Define the function class $\ell_\mathcal{F} = \{\ell(f(\boldsymbol{x}), y)|f \in \mathcal{F}\}$, we have the following generalization bound.

**Proposition 1.** *(Mohri et al. (2018); Bartlett & Mendelson (2002)) Suppose that the range of the loss function $\ell(f(x), y)$ is $[0, C]$. Then, for any $\delta \in (0, 1)$, with probability at least $1 - \delta$, the following holds for all $f \in \mathcal{F}$,*

$$R(f) \leq R_m(f) + 2C\mathcal{R}_\mathcal{S}(\ell_\mathcal{F}) + 3C\sqrt{\frac{\log \frac{2}{\delta}}{2m}}.$$

## 2.2 ROBUST GENERALIZATION GAP AND ADVERSARIAL RADEMACHER COMPLEXITY

**Robust Generalization Gap.** Let $\tilde{\ell}(f(x), y) := \max_{\|x'-x\|_p \leq \epsilon} \ell(f(x'), y)$ be the adversarial loss. The adversarial population risk and the adversarial empirical risk are

$$\tilde{R}(f) = \mathbb{E}_{(x,y)\sim\mathcal{D}} \max_{\|x'-x\|_p \leq \epsilon} \ell(f(x'), y) \quad and \quad \tilde{R}_m(f) = \frac{1}{m}\sum_{i=1}^{m} \max_{\|x'-x\|_p \leq \epsilon} \ell(f(x'_i), y_i),$$

respectively. In this paper we consider general $\ell_p$ attacks for $p \geq 1$. The robust generalization gap is defined as follow:

$$\text{Robust Generalization Gap} := \tilde{R}(f) - \tilde{R}_m(f).$$

Let the adversarial hypothesis class be $\tilde{\ell}_\mathcal{F} = \{\tilde{\ell}(f(x), y) | f \in \mathcal{F}\}$, according to Proposition 1, we have the following adversarial generalization bound.

**Proposition 2.** *(Yin et al. (2019)) Suppose that the range of the loss function $\tilde{\ell}(f(x), y)$ is $[0, C]$. Then, for any $\delta \in (0, 1)$, with probability at least $1 - \delta$, the following holds for all $f \in \mathcal{F}$,*

$$\tilde{R}(f) \leq \tilde{R}_m(f) + 2C\mathcal{R}_\mathcal{S}(\tilde{\ell}_\mathcal{F}) + 3C\sqrt{\frac{\log \frac{2}{\delta}}{2m}}.$$

**Binary Classification.** We first discuss the binary classification case, then we discuss the extension to the multi-class classification case in section 5. Following (Yin et al. (2019); Awasthi et al. (2020)), we assume that the loss function can be written as $\ell(f(\boldsymbol{x}), y) = \phi(yf(\boldsymbol{x}))$ where $\phi$ is a non-increasing function. Then

$$\max_{\boldsymbol{x}'} \ell(f(\boldsymbol{x}'), y) = \phi(\min_{\boldsymbol{x}'} yf(\boldsymbol{x}')).$$

Assume that the function $\phi$ is $L_\phi$-Lipschitz, by Talagrand's Lemma (Ledoux & Talagrand (2013)), we have $\mathcal{R}_\mathcal{S}(\tilde{\ell}_\mathcal{F}) \leq L_\phi\mathcal{R}_\mathcal{S}(\tilde{\mathcal{F}})$, where we define the adversarial function class as

$$\tilde{\mathcal{F}} = \{\tilde{f} : (\boldsymbol{x}, y) \to \inf_{\|\boldsymbol{x}-\boldsymbol{x}'\|_p \leq \epsilon} yf(\boldsymbol{x}') | f \in \mathcal{F}\}. \tag{1}$$

**Adversarial Rademacher Complexity.** We define $\mathcal{R}_\mathcal{S}(\tilde{\mathcal{F}})$ as adversarial Rademacher complexity. Our goal is to give upper bounds for adversarial Rademacher complexity. Then, it induces the guarantee of the robust generalization gap.

**Hypothesis Class.** We consider depth-$d$, width-$h$ fully-connected neural networks,

$$\mathcal{F} = \{\boldsymbol{x} \to W_d\rho(W_{d-1}\rho(\cdots\rho(W_1\boldsymbol{x})\cdots)), \|W_l\| \leq M_l, l = 1\cdots, d\}, \tag{2}$$

where $\rho(\cdot)$ is an element-wise $L_\rho$-Lipschitz activation function, $W_l$ are $h_l \times h_{l-1}$ matrices, for $l = 1, \cdots, d$. We have $h_d = 1$ and $h_0 = q$ is the dimension of the data $\boldsymbol{x}$. Let $h = \max\{h_0, \cdots, h_d\}$ be the width of the neural networks. Convolution neural networks are also included in this hypothesis class because convolution layer can be viewed as a special form of fully-connected layer. Denote the $(a, b)$-group norm $\|W\|_{a,b}$ as the $a$-norm of the $b$-norm of the rows of $W$. We consider two cases, Frobenius norm and $\|\cdot\|_{1,\infty}$-norm in equation (2). Additionally, we assume that $\|\boldsymbol{X}\|_{p,\infty} = B$.

## 2.3 RADEMACHER COMPLEXITY AND COVERING NUMBER

**Covering Number.** Our solution for adversarial Rademacher complexity is based on the covering number. We first provide the definition of covering number.

**Definition 1** ($\varepsilon$-cover). *Let $\varepsilon > 0$ and $(V, d(\cdot, \cdot))$ be a metric space, where $d(\cdot, \cdot)$ is a (pseudo)-metric. $\mathcal{C} \subset V$ is an $\varepsilon$-cover of $V$, if for any $v \in V$, there exists $v' \in \mathcal{C}$ s.t. $d(v, v') \leq \varepsilon$. Define the smallest $|\mathcal{C}|$ as $\varepsilon$-covering number of $V$ and denote as $\mathcal{N}(V, d(\cdot, \cdot), \varepsilon)$.*

Next, we define the $\varepsilon$-covering number of a function class $\mathcal{F}$. Given the sample dataset $S = \{(\boldsymbol{x}_1, y_1), \cdots, (\boldsymbol{x}_m, y_m)\}$ with $\boldsymbol{x}_i \in \mathbb{R}^d$, let $\|f\|_S^2 = \frac{1}{m} \sum_{i=1}^m f(\boldsymbol{x}_i, y_i)^2$ be a pseudometric of $\mathcal{F}$. Define the $\varepsilon$-covering number of $\mathcal{F}$ be $\mathcal{N}(\mathcal{F}, \|\cdot\|_S, \varepsilon)$. Let $D$ be the diameter of $\mathcal{F}$ with $D = 2 \max_{f \in \mathcal{F}} \|f\|_S$.

**Proposition 3** (Dudley's integral). *The Rademacher complexity $\mathcal{R}(\mathcal{F})$ satisfiy*

$$\mathcal{R}_S(\mathcal{F}) \leq \frac{12}{\sqrt{m}} \int_0^{D/2} \sqrt{\log \mathcal{N}(\mathcal{F}, \|\cdot\|_S, \varepsilon)} d\varepsilon.$$

The proof of Dudley's integral can be found in statistic textbooks (e.g. (Wainwright (2019))). Based on this, we can bound the covering number of the function class $\mathcal{F}$ to give an upper bound of the Rademacher complexity.

## 3 RELATED WORK

**Adversarial Attacks and Defense.** Starting from the work of (Szegedy et al. (2013)), it has now been well known that deep neural networks trained via standard gradient descent based algorithms are highly susceptible to imperceptible corruptions to the input data (Goodfellow et al. (2014); Chen et al. (2017); Carlini & Wagner (2017); Madry et al. (2017)). This has led to a series of work aimed at training neural networks robust to such perturbations (Wu et al. (2020); Gowal et al. (2020); Wu et al. (2020)) and works aimed at designing more sophisticated attacks to attack the classifiers (Athalye et al. (2018); Tramer et al. (2020); Chen et al. (2017)).

**Adversarial Generalization.** The work of (Schmidt et al. (2018); Raghunathan et al. (2019); Zhai et al. (2019)) have shown that in some scenarios achieving adversarial generalization requires more data. The work of (Attias et al. (2021); Montasser et al. (2019)) explains generalization in adversarial settings using VC-dimension. Cullina et al. (2018) studies PAC-learning guarantees in the adversarial setting via VC-dimension. VC-dimension usually depends on the number of parameters in the model, while Rademacher complexity usually depends on the weight matrices. Rademacher complexity usually provides tighter generalization bounds (Bartlett (1998)). Neyshabur et al. (2017b) uses a pac-bayesian approach to provide a generalization bound for neural networks. Sinha et al. (2017) study the generalization of an adversarial training algorithm in terms of distributional robustness. The work of (Xing et al. (2021a;b); Javanmard et al. (2020)) study the generalization properties in the setting of linear regression. Gaussian mixture models are used to analyze adversarial generalization (Taheri et al. (2020); Javanmard et al. (2020); Dan et al. (2020)). The work of (Allen-Zhu & Li (2020)) explains adversarial generalization through the lens of feature purification.

**Adversarial Rademacher Complexity.** Researchers have analyzed adversarial Rademacher complexity in linear and two-layers neural networks cases. In linear cases, the upper bounds can be directly derived by definition (Khim & Loh (2018); Yin et al. (2019)). In two-layers neural networks cases, an upper bound is derived using Massart's Lemma (Awasthi et al. (2020)). It seems that these proofs cannot be extended to multi-layers cases. Moreover, based on the definition of adversarial function class $\tilde{\mathcal{F}}$ in equation (1), the candidate functions are not composition functions, but with an $\inf$ operation in front of the neural networks. Then, the induction on layers seems not applicable in calculating adversarial Rademacher complexity for deep neural networks. The works of (Khim & Loh (2018)) and (Gao & Wang (2021)) indicate the difficulty of analyzing adversarial Rademacher complexity. They analyze other variants of adversarial Rademacher complexity of DNNs. The first one introduce tree transformation, but it overestimates the adversarial loss. The second one considers fast gradient sign methods (FGSM) adversarial examples, but it also requires additional assumption on the gradient.

In Appendix B, we provide the details of the above bounds and discuss why these methods seem not applicable in multi-layers cases. We also provide a comparison of adversarial generalization in Rademacher complexity framework and other frameworks. In the next section, we provide our solution of adversarial Rademacher complexity based on covering number and analyze each factor in the upper bound.

## 4 OUR SOLUTION OF ADVERSARIAL RADEMACHER COMPLEXITY

The following Theorem states an upper bound of adversarial Rademacher complexity in the Frobenius norm cases.

**Theorem 1** (Frobenius Norm Bound). *Given the function class $\mathcal{F}$ in equation (2) under Frobbenius Norm, and the corresponding adversarial function class $\tilde{\mathcal{F}}$ in equation (1). The adversarial Rademacher complexity of deep neural networks $\mathcal{R}_S(\tilde{\mathcal{F}})$ satisfies*

$$\mathcal{R}_S(\tilde{\mathcal{F}}) \leq \frac{24}{\sqrt{m}} \max\{1, q^{\frac{1}{2}-\frac{1}{p}}\}(\|\boldsymbol{X}\|_{p,\infty} + \epsilon)L_\rho^{d-1}\sqrt{\sum_{l=1}^d h_l h_{l-1} \log(3d)} \prod_{l=1}^d M_l. \tag{3}$$

*By assuming that $L_\rho = 1$, $p \leq 2$, $\|\boldsymbol{X}\|_{p,\infty} = B$, and $h = \max\{h_0, \cdots, h_d\}$, we have*

$$\mathcal{R}_S(\tilde{\mathcal{F}}) \leq \mathcal{O}\bigg(\frac{(B+\epsilon)h\sqrt{d\log(d)}\prod_{l=1}^d M_l}{\sqrt{m}}\bigg). \tag{4}$$

Because of the $\inf$ operation of the function in $\tilde{\mathcal{F}}$, we cannot peel off the layers or calculate the covering number by induction on layers. Our proof is based on calculating the covering number of $\tilde{\mathcal{F}}$ directly. Below we sketch the proof. The completed proof is provided in Appendix A.

**Step 1: Diameter of $\tilde{\mathcal{F}}$.** We first calculate the diameter of $\tilde{\mathcal{F}}$. We have

$$2\max_{\tilde{f}\in\tilde{\mathcal{F}}}\|\tilde{f}\|_S \leq 2L_\rho^{d-1}\max\{1, q^{\frac{1}{2}-\frac{1}{p}}\}(\|\boldsymbol{X}\|_{p,\infty}+\epsilon)\prod_{l=1}^d M_l \overset{\Delta}{=} D.$$

**Step 2: Distance to $\tilde{\mathcal{F}}^c$.** Let $\mathcal{C}_l$ be $\delta_l$-covers of $\{\|W_l\|_F \leq M_l\}$, $l = 1, 2, \cdots, d$. Let

$$\mathcal{F}^c = \{f^c : \boldsymbol{x} \to W_d^c \rho(W_{d-1}^c \rho(\cdots \rho(W_1^c \boldsymbol{x})\cdots)), W_l^c \in \mathcal{C}_l, l = 1, 2 \cdots, d\}$$

$$\text{and } \tilde{\mathcal{F}}^c = \{\tilde{f} : (\boldsymbol{x}, y) \to \inf_{\|\boldsymbol{x}-\boldsymbol{x}'\|_p \leq \epsilon} yf(\boldsymbol{x}')|f \in \mathcal{F}^c\}.$$

For all $\tilde{f} \in \tilde{\mathcal{F}}$, we need to find the smallest distance to $\tilde{\mathcal{F}}^c$, i.e. we need to calculate the

$$\max_{\tilde{f}\in\tilde{\mathcal{F}}} \min_{\tilde{f}^c\in\tilde{\mathcal{F}}^c} \|\tilde{f} - \tilde{f}^c\|_S.$$

$\forall (\boldsymbol{x}, y) \in \mathcal{D}$, given $f$ and $f^c$, let $\boldsymbol{x}^* = \arg\inf_{\|\boldsymbol{x}-\boldsymbol{x}'\|_p} yf(\boldsymbol{x}')$ and $\boldsymbol{x}^c = \arg\inf_{\|\boldsymbol{x}-\boldsymbol{x}'\|_p} yf^c(\boldsymbol{x}')$.
Let $z = \begin{cases} \boldsymbol{x}^c & if \quad f(\boldsymbol{x}^*) \geq f^c(\boldsymbol{x}^c) \\ \boldsymbol{x} & if \quad f(\boldsymbol{x}^*) < f^c(\boldsymbol{x}^c) \end{cases}$ and $g_b^a(z) = W_b\rho(\cdots W_{a+1}\rho(W_a^c \cdots \rho(W_1^c z)\cdots)))$. Then

$$|f(z) - f^c(z)| = |g_d^0(z) - g_d^d(z)| \leq |g_d^0(z) - g_d^1(z)| + \cdots + |g_d^{d-1}(z) - g_d^d(z)| \leq \sum_{l=1}^d \frac{D\delta_l}{2M_l}.$$

Let $\delta_l = 2M_l \varepsilon/dD$, $l = 1, \cdots, d$, we have $\max_{\tilde{f}\in\tilde{\mathcal{F}}} \min_{\tilde{f}^c\in\tilde{\mathcal{F}}^c} \|\tilde{f} - \tilde{f}^c\|_S \leq \sum_{l=1}^d \frac{D\delta_l}{2M_l} \leq \varepsilon$.

**Step 3: Covering Number of $\tilde{\mathcal{F}}$.** Then, We can calculate the $\varepsilon$-covering number $\mathcal{N}(\tilde{\mathcal{F}}, \|\cdot\|_S, \varepsilon)$.
Because $\tilde{\mathcal{F}}^c$ is a $\varepsilon$-cover of $\tilde{\mathcal{F}}$. The cardinality of $\tilde{\mathcal{F}}^c$ is

$$\mathcal{N}(\tilde{\mathcal{F}}, \|\cdot\|_S, \varepsilon) = |\tilde{\mathcal{F}}^c| \leq (\frac{3dD}{2\varepsilon})^{\sum_{l=1}^d h_l h_{l-1}}. \tag{5}$$

**Step 4: Integration.** By Dudley's integral, we obtain the bound in Theorem 1. □

Remark: Step 2 is the critical step in the proof. In words, if we calculate the covering number of the class of $d$-layers neural nets directly, we only need to define the optimal adversarial example one time. Then we can calculate the other things using this adversarial example. In contrast, if we want to do it layer by layer, the optimal adversarial example will be changed when we add or peel off a layer. This is why the induction methods fail in adversarial settings.

**Theorem 2** ($\|\cdot\|_{1,\infty}$-Norm Bound). *Given the function class $\mathcal{F}$ in equation (2) under $\|\cdot\|_{1,\infty}$-norm, and the corresponding adversarial function class $\tilde{\mathcal{F}}$ in equation (1). The adversarial Rademacher complexity of deep neural networks $\mathcal{R}_S(\tilde{\mathcal{F}})$ satisfies*

$$\mathcal{R}_S(\tilde{\mathcal{F}}) \leq \frac{24}{\sqrt{m}}(\|\boldsymbol{X}\|_{p,\infty} + \epsilon)L_\rho^{d-1}\sqrt{\sum_{l=1}^{d} h_l h_{l-1} \log(3d)}\prod_{l=1}^{d} M_l. \tag{6}$$

In the case of $\|\cdot\|_{1,\infty}$-norm, the bound is similar to the bound in the Frobenius norm case except the term $\max\{1, q^{1/2-1/p}\}$. Therefore, for all $p \geq 1$, the $\|\cdot\|_{1,\infty}$-norm bound have the same order in equation (4).

**Theorem 3** (Lower Bound). *Given the function class of DNNs $\mathcal{F}$ in equation (2), and the corresponding adversarial function class $\tilde{\mathcal{F}}$ in equation (1). Exist sample dataset S, s.t. the adversarial Rademacher complexity of deep neural networks $\mathcal{R}_S(\tilde{\mathcal{F}})$ satisfies*

$$\mathcal{R}_S(\tilde{\mathcal{F}}) \geq \Omega\left(\frac{(B+\epsilon)\prod_{l=1}^{d} M_l}{\sqrt{m}}\right). \tag{7}$$

The proof of the above Theorem is based on constructing a scalar network and is provided in Appendix A. The gap between the upper bound and the lower bound is the dependence on depth-$d$ and width-$h$, $h\sqrt{d\log d}$. In the next section, we extend the adversarial Rademacher complexity to the Multi-class classification cases.

## 5 MARGIN BOUNDS FOR MULTI-CLASS CLASSIFICATION

### 5.1 SETTING FOR MULTI-CLASS CLASSIFICATION

The setting for multi-class classification follows (Bartlett & Mendelson (2002)). In a K-class classification problem, let $\mathcal{Y} = \{1, 2, \cdots, K\}$. The functions in the hypothesis class $\mathcal{F}$ map $\mathcal{X}$ to $\mathbb{R}^K$, the $k$-th output of $f$ is the score of $f(\boldsymbol{x})$ assigned to the $k$-th class.

Define the margin operator $M(f(\boldsymbol{x}), y) = [f(\boldsymbol{x})]_y - \max_{y' \neq y}[f(\boldsymbol{x})]_{y'}$. The function makes a correct prediction if and only if $M(f(\boldsymbol{x}), y) > 0$. We consider a particular loss function $\ell(f(\boldsymbol{x}), y) = \phi_\gamma(M(f(\boldsymbol{x}), y))$, where $\gamma > 0$ and $\phi_\gamma : \mathbb{R} \to [0, 1]$ is the ramp loss:

$$\phi_\gamma(t) = \begin{cases} 1 & t \leq 0 \\ 1 - \frac{t}{\gamma} & 0 < t < \gamma \\ 0 & t \geq \gamma. \end{cases}$$

The loss function $\ell(f(\boldsymbol{x}), y)$ satisfies:

$$\mathbb{1}(y \neq \arg\max_{y' \in [K]}[f(\boldsymbol{x})]_{y'}) \leq \ell(f(\boldsymbol{x}), y) \leq \mathbb{1}([f(\boldsymbol{x})]_y \leq \gamma + \max_{y' \neq y}[f(\boldsymbol{x})]_{y'}). \tag{8}$$

Define the function class $\ell_{\mathcal{F}} := \{(\boldsymbol{x}, y) \mapsto \phi_\gamma(M(f(\boldsymbol{x}), y)) : f \in \mathcal{F}\}$. Since $\phi_\gamma(t) \in [0, 1]$ and $\phi_\gamma(\cdot)$ is $1/\gamma$-Lipschitz, by combining (8) with Theorem 1, we can obtain the following direct corollary as the generalization bound in the multi-class classification.

**Corollary 1** (Mohri et al. (2018)). *Consider the above multi-class classification setting. For any fixed $\gamma > 0$, we have with probability at least $1 - \delta$, for all $f \in \mathcal{F}$,*

$$\mathbb{P}_{(\boldsymbol{x}, y) \sim \mathcal{D}}\left\{y \neq \arg\max_{y' \in [K]}[f(\boldsymbol{x})]_{y'}\right\}$$

$$\leq \frac{1}{m}\sum_{i=1}^{m}\mathbb{1}([f(\boldsymbol{x}_i)]_{y_i} \leq \gamma + \max_{y' \neq y}[f(\boldsymbol{x}_i)]_{y'}) + 2\mathcal{R}_S(\ell_{\mathcal{F}}) + 3\sqrt{\frac{\log\frac{2}{\delta}}{2m}}.$$

In adversarial training, let $\mathbb{B}_{\boldsymbol{x}}^p(\epsilon) = \{x' : \|\boldsymbol{x}' - \boldsymbol{x}\|_p \leq \epsilon\}$ and we define the adversarial function class $\tilde{\ell}_{\mathcal{F}} := \{(\boldsymbol{x}, y) \mapsto \max_{x' \in \mathbb{B}_{\boldsymbol{x}}^p(\epsilon)} \ell(f(\boldsymbol{x}'), y) : f \in \mathcal{F}\}$. We have

**Corollary 2** (Yin et al. (2019)). *Consider the above adversarial multi-class classification setting. For any fixed $\gamma > 0$, we have with probability at least $1 - \delta$, for all $f \in \mathcal{F}$,*

$$\mathbb{P}_{(\boldsymbol{x},y) \sim \mathcal{D}} \left\{ \exists \ \boldsymbol{x}' \in \mathbb{B}_{\boldsymbol{x}}^p(\epsilon) \ s.t. \ y \neq \arg \max_{y' \in [K]} [f(\boldsymbol{x}')]_{y'} \right\}$$

$$\leq \frac{1}{m} \sum_{i=1}^{m} \mathbb{1}(\exists \ \boldsymbol{x}_i' \in \mathbb{B}_{\boldsymbol{x}_i}^p(\epsilon) \ s.t. \ [f(\boldsymbol{x}_i')]_{y_i} \leq \gamma + \max_{y' \neq y}[f(\boldsymbol{x}_i')]_{y'}) + 2\mathcal{R}_S(\tilde{\ell}_{\mathcal{F}}) + 3\sqrt{\frac{\log \frac{2}{\delta}}{2m}}.$$

## 5.2 ADVERSARIAL RADEMACHER COMPLEXITY

Under the multi-class setting, we have the following bound for adversarial Rademacher complexity.

**Theorem 4.** *Given the function class $\mathcal{F}$ in equation (2) under Frobbenius Norm, and the corresponding adversarial function class $\tilde{\mathcal{F}}$ in equation (1). The adversarial Rademacher complexity of deep neural networks $\mathcal{R}_S(\tilde{\ell}_{\mathcal{F}})$ satisfies*

$$\mathcal{R}_S(\tilde{\ell}_{\mathcal{F}}) \leq \frac{48K}{\gamma\sqrt{m}} \max\{1, q^{\frac{1}{2} - \frac{1}{p}}\}(\|\boldsymbol{X}\|_{p,\infty} + \epsilon)L_\rho^{d-1}\sqrt{\sum_{l=1}^{d} h_l h_{l-1} \log(3d)} \prod_{l=1}^{d} M_l. \tag{9}$$

The $\|\cdot\|_{1,\infty}$-norm bound is similar, except the term $\max\{1, q^{\frac{1}{2} - \frac{1}{p}}\}$. Below we sketch the proof. Step 1: Let $\tilde{\mathcal{F}}^k = \{(\boldsymbol{x}, y) \to \inf_{\|\boldsymbol{x}' - \boldsymbol{x}\| \leq \epsilon}([f(\boldsymbol{x}')]_y - [f(\boldsymbol{x}')]_k), f \in \mathcal{F}\}$, then $\mathcal{R}_S(\tilde{\ell}_{\mathcal{F}}) \leq K\mathcal{R}_S(\tilde{\ell}_{\mathcal{F}^k})$. Step 2: By the Lipschitz property of $\phi_\gamma(\cdot)$, $\mathcal{R}_S(\tilde{\ell}_{\mathcal{F}^k}) \leq \frac{1}{\gamma}\mathcal{R}_S(\mathcal{F}^k)$. Step 3: The calculation of $\mathcal{R}_S(\tilde{\mathcal{F}}^k)$ follows the binary case.

## 5.3 COMPARISON OF THE BOUNDS

Now, we compare the difference between the bounds for (standard) Rademacher complexity and adversarial Rademacher complexity. We have shown that

$$\mathcal{R}_S(\ell_{\mathcal{F}}) \leq \mathcal{O}\left(\frac{B\sqrt{d}\prod_{l=1}^{d} M_l}{\gamma\sqrt{m}}\right) \text{ and } \mathcal{R}_S(\tilde{\ell}_{\mathcal{F}}) \leq \mathcal{O}\left(\frac{(B+\epsilon)h\sqrt{d\log d}\prod_{l=1}^{d} M_l}{\gamma\sqrt{m}}\right), \tag{10}$$

where we use the upper bound of $\mathcal{R}_S(\mathcal{F})$ in (Golowich et al. (2018)).

**Algorithm-Independent Factors.** In the two bounds, the algorithm-independent factors include Sample size $B$, perturbation intensity $\epsilon$, depth-$d$, and width-$h$. To simplify the notations, we let $C_{std} = B\sqrt{d}$ and $C_{adv} = (B+\epsilon)h\sqrt{d\log d}$ be the constants in standard and adversarial Rademacher complexity, respectively. We simply have $C_{adv} > C_{std}$.

**Algorithm-Dependent Factors.** In the two bounds, the margins $\gamma$ and the product of the matrix norms $\prod_{l=1}^{d} \|W_l\|$ depend on the training algorithms. To simplify the notations, we define $W_{std} := \prod_{l=1}^{d} \|W_l\|/\gamma$ if the training algorithm is standard training. Correspondingly, let $W_{adv} := \prod_{l=1}^{d} \|W_l\|/\gamma$ if the training algorithm is adversarial training. In the next section, we conduct experiments to show that $W_{adv} > W_{std}$.

**Notation of generalization gaps.** In the next section, we use $\mathcal{E}(\cdot)$ and $\tilde{\mathcal{E}}(\cdot)$ to denote the standard and robust generalization gap. We use $f_{std}$ and $f_{adv}$ to denote the standard- and adversarially-trained model. Our goal is to understand why the robust generalization gap of an adversarial training model is large, which is quite different from the standard generalization gap of a standard-trained model, i.e., we want to analyze why $\tilde{\mathcal{E}}(f_{adv}) > \mathcal{E}(f_{std})$. Based on the standard and adversarial Rademacher complexity bounds, it is suggested that

$$\tilde{\mathcal{E}}(f_{adv}) \propto C_{adv}W_{adv} \quad and \quad \mathcal{E}(f_{std}) \propto C_{std}W_{std}.$$

To analyze the individual effect of factors $C_{adv}$ and $W_{adv}$, we further introduce two kinds of generalization gaps, the robust generalization gap of a standard-trained model ($\tilde{\mathcal{E}}(f_{std})$) and the standard generalization gap of an adversarially-trained model ($\mathcal{E}(f_{adv})$). The standard and adversarial Rademacher complexity suggest that

$$\tilde{\mathcal{E}}(f_{std}) \propto C_{adv}W_{std} \quad and \quad \mathcal{E}(f_{adv}) \propto C_{std}W_{adv}.$$

## 6 EXPERIMENT

As we discuss in the previous section, the product of weight norm $\prod_{l=1}^{d} \|W_l\|$ and the margin $\gamma$ are algorithm-dependent factors controlling the generalization gap. We provide experiments comparing the difference between these terms in standard and adversarial settings. Since the bounds also hold for convolution neural networks, we consider the experiments of training VGG networks (Simonyan & Zisserman (2014)) on CIFAR-10 (Krizhevsky et al. (2009)). We use the experiments on VGG-19 to illustrate the results. Other experiments are provided in Appendix C.

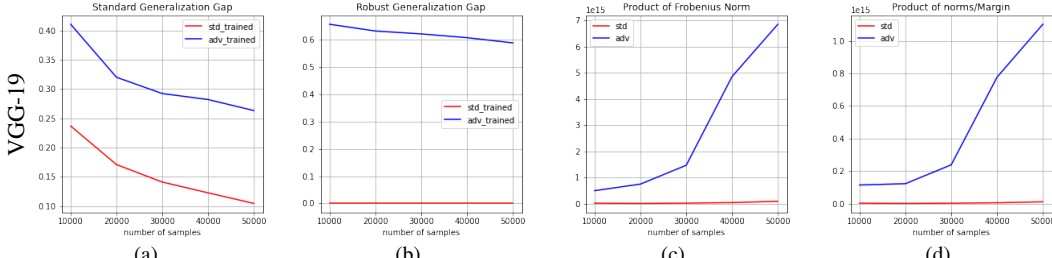

Figure 1: Product of the Frobenius norm in the experiments on CIFAR-10. The red lines are the results of standard training. The blue lines are the results of adversarial training. (a): Standard Generalization gap, the blue line represents $\mathcal{E}(f_{adv})$ and the red line represents $\mathcal{E}(f_{std})$. (b): Robust Generalization Gap, the blue line represents $\tilde{\mathcal{E}}(f_{adv})$ and the red line represents $\tilde{\mathcal{E}}(f_{std})$. (c): $\prod_{l=1}^{d} \|W_l\|_F$ of the neural networks. (d): $\prod_{l=1}^{d} \|W_l\|_F/\gamma$ of the neural networks, the blue line represents $W_{adv}$ and the red line represents $W_{std}$.

**Training Settings.** For both standard and adversarial training, we use the stochastic gradient descent (SGD) optimizer, along with a learning rate schedule, which is 0.1 over the first 100 epochs, down to 0.01 over the following 50 epochs, and finally be 0.001 in the last 50 epochs. For adversarial settings, we adopt the $\ell_\infty$ PGD adversarial training (Madry et al. (2017)). The perturbation intensity is set to be $8/255$. We set the number of steps as 20 and further increase it to 40 in the testing phase. For the stepsize in the inner maximization, we set it as 2/255. In Corollary 2, we need to use the optimal adversarial examples to calculate the margin and the robust generalization gap, but it is unknown in practice. We use the PGD adversarial examples as substitutes.

**Calculation of Margins.** We adopt the setting in (Neyshabur et al. (2017a)). In standard training, we set the margin over training set to be $5^{th}$-percentile of the margins of the data points in $S$. i.e. $\text{Prc}_5\{f(\boldsymbol{x}_i)[y_i] - \max_{y \neq y_i} f(\boldsymbol{x})[y]|(\boldsymbol{x}_i, y_i) \in S\}$. In adversarial settings, we set the margin over training set to be $5^{th}$-percentile of the margins of the PGD-adversarial examples of $S$. The choice of $5^{th}$-percentile is because the training accuracy is $100\%$ in all the experiments. We provide ablation studies about the percentile in the Appendix C.

**Standard and Robust Generalization Gap.** In Figure 3 (a) and (b), we plot the standard and robust generalization gap of both standard-trained and adversarially-trained models. We use the results using 50000 training samples to discuss the experiments. Firstly, in Figure 3 (a), we can see that $\mathcal{E}(f_{std})$ is small (=10.45%). On the other hand, an adversarial-trained model has a larger standard generalization gap ($\mathcal{E}(f_{adv})$=26.34%). It is a widely observed phenomenon that adversarial training hurts standard generalization. One reason is that adversarial training overfits the adversarial examples and performs worse on the original examples. Secondly, in Figure 3 (b), the robust generalization gap of a standard-trained model is very small ($\tilde{\mathcal{E}}(f_{std}) = 0$). It is because the standard-trained model can easily be attacked on both the training set and test set. Then, the robust training accuracy and test accuracy are closed to 0%. Therefore, the robust generalization gap is also 0. On the contrary, $\tilde{\mathcal{E}}(f_{adv})$=58.90%, i.e. the adversarial generalization is bad. This is also observed in the previous studies, and we aim to discuss the reasons.

**Adversarially-trained Models Have Larger Weight Norms, i.e. $W_{adv} > W_{std}$.** In Figure 3 (c), we can see that the $\prod_{l=1}^{d} \|W_l\|_F$ of adversarial training is much larger than the $\prod_{l=1}^{d} \|W_l\|_F$ of standard training. In (d), the $\prod_{l=1}^{d} \|W_l\|_F$ is divided by $\gamma$, we can see that the Figures is similar to the Figures in (c), $W_{adv}$ is larger than $W_{std}$. One of the reasons why $W_{adv} > W_{std}$ is neural networks need more capacity to fit the adversarial examples.

Table 1: Comparison of the four kinds of generalization Gap introduced in Section 5.3. The experiments are training VGG-19 on CIFAR-10. Notice that $\tilde{\mathcal{E}}(f_{std})$=0% is a degenerated case, with training error=100%. In the other three cases, the training errors≈0%.

| | Standard-trained models | | Adversarially-trained models | |
|---|---|---|---|---|
| Types of Generalization Gaps | Standard | Robust | Standard | Robust |
| Training Errors | 0% | 100% | 0% | 0.02% |
| Test Errors | 10.45% | 100% | 26.34% | 58.92% |
| Generalization Gaps | $\mathcal{E}(f_{std})$=10.45% | $\tilde{\mathcal{E}}(f_{std})$=0% | $\mathcal{E}(f_{adv})$=26.34% | $\tilde{\mathcal{E}}(f_{adv})$=58.90% |

$\tilde{\mathcal{E}}(f_{std})$=0% **is a degenerated case.** In Table 1, we show the training and test errors for all kinds of generalization gaps. We can see that the robust training error for a standard-trained model is equal to 100%. Since the model does not fit any adversarial examples in the training set, there is nothing to generalize to the adversarial examples in the test set. The generalization gap becomes meaningless. And the Rademacher complexity bound $\tilde{\mathcal{E}}(f_{std}) \leq \mathcal{O}(C_{adv}W_{std}/\sqrt{m})$ becomes a trivial bound. In the other three cases, the training errors are all ≈0%. The generalization gaps are meaningful. We aim to analyze why $\tilde{\mathcal{E}}(f_{adv}) > \mathcal{E}(f_{std})$ by analyzing $\tilde{\mathcal{E}}(f_{adv}) > \mathcal{E}(f_{adv}) > \mathcal{E}(f_{std})$.

**The effect of $C_{adv}$.** We first compare the difference between $\tilde{\mathcal{E}}(f_{adv})$ and $\mathcal{E}(f_{adv})$. We can see that $\tilde{\mathcal{E}}(f_{adv}) = 58.90\% > \mathcal{E}(f_{adv}) = 26.34\%$. For an adversarially-trained model, the robust generalization gap is larger than the standard generalization gap. If we use the bounds of adversarial and standard Rademacher complexity as approximations of the robust and standard generalization gap, i.e., $\tilde{\mathcal{E}}(f_{adv}) \propto C_{adv}W_{adv}$ and $\mathcal{E}(f_{adv}) \propto C_{std}W_{adv}$, $\tilde{\mathcal{E}}(f_{adv}) > \mathcal{E}(f_{adv})$ can be explained by $C_{adv} > C_{std}$ since $W_{adv}$ are the same in the two bounds.

**The effect of $W_{adv}$.** Similarly, we compare the difference between $\mathcal{E}(f_{adv})$ and $\mathcal{E}(f_{std})$. We can see that $\mathcal{E}(f_{adv}) = 26.34\% > \mathcal{E}(f_{std}) = 10.45\%$. This is a widely observed phenomenon that adversarial training hurts standard generalization. It can also be explained by the Rademacher bounds. If we use the bounds of standard Rademacher complexity as approximations, i.e., $\mathcal{E}(f_{adv}) \propto C_{std}W_{adv}$ and $\mathcal{E}(f_{std}) \propto C_{std}W_{std}$, $\tilde{\mathcal{E}}(f_{adv}) > \mathcal{E}(f_{adv})$ can be explained by $W_{adv} > W_{std}$.

In summary, we can use a simple formula to explain why $\tilde{\mathcal{E}}(f_{adv}) > \mathcal{E}(f_{adv}) > \mathcal{E}(f_{std})$ through the lens of Rademacher complexity. That is

$$C_{adv}W_{adv} > C_{std}W_{adv} > C_{std}W_{std}.$$

The difficulty of adversarial generalization comes from two parts, the constant $C_{adv}$ and the weight norms $W_{adv}$. The first part $C_{adv}$ is independent of the algorithms. It comes from the minimax problem of adversarial training itself, and it cannot be avoided. The second part $W_{adv}$ depends on the algorithms. Therefore, the product of the weight norms is an important factor for the robust generalization of adversarial training.

**Ablation Studies.** We provide other ablation studies in Appendix C. First, we consider different VGG architecture and give the experiments on VGG-11, 13, and 16. Secondly, We consider different percentile of the margins of the training dataset. Thirdly, we provide the experiments on $\| \cdot \|_{1,\infty}$-norm. We can see that the $\| \cdot \|_{1,\infty}$-norm bound has a larger magnitude than the Frobenius norm bound. Then, we provide the experiments on CIFAR-100. Finally, the large weight norms suggest adding a regularization term on the weights during training. We provide experiments with and without weight decay and see that the one with weight decay has a smaller generalization gap and a smaller $\prod_{l=1}^{d} \|W_l\|_F / \gamma$. These experiments suggest the strong relationship between robust generalization gap and the product of weight norms.

## 7 CONCLUSION

In this paper, we first provide upper bounds for the adversarial Rademacher complexity of deep neural networks. Then, we experimentally investigate these bounds and show that the product of weight norms is a key factor explaining why adversarial training cannot generalize well. We think our results will motivate more theoretical research to understand adversarial training and empirical research to improve the generalization of adversarial training.

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

# A  PROOF OF THE THEOREMS

## A.1  PROOF OF THEOREM 1

**Theorem 1** (Frobenius Norm Bound). *Given the function class $\mathcal{F}$ in equation (2) under Frobbenius Norm, and the corresponding adversarial function class $\tilde{\mathcal{F}}$ in equation (1). The adversarial Rademacher complexity of deep neural networks $\mathcal{R}_S(\tilde{\mathcal{F}})$ satisfies*

$$\mathcal{R}_S(\tilde{\mathcal{F}}) \leq \frac{24}{\sqrt{m}} \max\{1, q^{\frac{1}{2}-\frac{1}{p}}\}(\|\boldsymbol{X}\|_{p,\infty} + \epsilon)L_\rho^{d-1}\sqrt{\sum_{l=1}^{d} h_l h_{l-1} \log(3d)} \prod_{l=1}^{d} M_l.$$

*By assuming that $L_\rho = 1$, $p \leq 2$, $\|\boldsymbol{X}\|_{p,\infty} \leq B$, and $h = \max\{h_0, \cdots, h_d\}$, we have*

$$\mathcal{R}_S(\tilde{\mathcal{F}}) \leq \mathcal{O}\left(\frac{(B+\epsilon)h\sqrt{d\log(d)}\prod_{l=1}^{d} M_l}{\sqrt{m}}\right).$$

Because of the $\inf$ operation of the function in $\tilde{\mathcal{F}}$, we cannot peel off the layers or calculate the covering number by induction on layers. Our proof is based on calculating the covering number of $\tilde{\mathcal{F}}$ directly. Before we provide the proof, we first introduce the following lemma.

**Lemma 1** (Covering number of norm-balls). *Let $\mathcal{B}$ be a $\ell_p$ norm ball with radius $W$. Let $d(\boldsymbol{x}_1, \boldsymbol{x}_2) = \|\boldsymbol{x}_1 - \boldsymbol{x}_2\|_p$. Define the $\varepsilon$-covering number of $B$ as $\mathcal{N}(\mathcal{B}, d(\cdot, \cdot), \varepsilon)$, we have*

$$\mathcal{N}(\mathcal{B}, d(\cdot, \cdot), \varepsilon) \leq (1 + 2W/\varepsilon)^q.$$

In the case of Frobenius norm ball of $m \times n$ matrices, we have the dimension $q = m \times n$ and

$$\mathcal{N}(\mathcal{B}, \|\cdot\|_F, \varepsilon) \leq (1 + 2W/\varepsilon)^{m \times n} \leq (3W/\varepsilon)^{m \times n}.$$

**Lemma 2.** *if $\boldsymbol{x}_i^* \in \{\boldsymbol{x}_i'|\|\boldsymbol{x}_i - \boldsymbol{x}_i'\|_p \leq \epsilon\}$, we have*

$$\|\boldsymbol{x}_i^*\|_{r^*} \leq \max\{1, q^{1-\frac{1}{r}-\frac{1}{p}}\}(\|\boldsymbol{X}\|_{p,\infty} + \epsilon).$$

Proof: If $p \geq r^*$, by Holder's inequality with $1/r^* = 1/p + 1/s$,

$$\|\boldsymbol{x}_i^*\|_{r^*} \leq \sup \|\mathbf{1}\|_s \|\boldsymbol{x}_i^*\|_p = \|\mathbf{1}\|_s \|\boldsymbol{x}_i^*\|_p = q^{\frac{1}{s}}\|\boldsymbol{x}_i^*\|_p = q^{1-\frac{1}{r}-\frac{1}{p}}\|\boldsymbol{x}_i^*\|_p.$$

Equality holds when all the entries are equal. If $p < r^*$, we have

$$\|\boldsymbol{x}_i^*\|_{r^*} \leq \|\boldsymbol{x}_i^*\|_p.$$

Equality holds when one of the entries of $\theta$ equals to one and the others equal to zero. Then

$$\begin{aligned}\|\boldsymbol{x}_i^*\|_{r^*} &\leq \max\{1, q^{1-\frac{1}{r}-\frac{1}{p}}\}\|\boldsymbol{x}_i^*\|_p \\ &\leq \max\{1, q^{1-\frac{1}{r}-\frac{1}{p}}\}(\|\boldsymbol{x}_i\|_p + \|\boldsymbol{x}_i - \boldsymbol{x}_i^*\|_p) \\ &\leq \max\{1, q^{1-\frac{1}{r}-\frac{1}{p}}\}(\|\boldsymbol{X}\|_{p,\infty} + \epsilon).\end{aligned}$$

$\square$

**Lemma 3.** *Let $A$ be a $m \times n$ matrix and $b$ be a $n$-dimension vector, we have*

$$\|A \cdot b\|_2 \leq \|A\|_F \|b\|_2.$$

Proof: let $A_i$ be the rows of $A$, $i = 1 \cdots m$, we have

$$\|A \cdot b\|_2 = \sqrt{\sum_{i=1}^{m}(A_i b)^2} \leq \sqrt{\sum_{i=1}^{m}\|A_i\|_2^2\|b\|_2^2} = \sqrt{\sum_{i=1}^{m}\|A_i\|_2^2}\sqrt{\|b\|_2^2} = \|A\|_F\|b\|_2.$$

$\square$

**Step 1: Diameter of $\tilde{\mathcal{F}}$.** We first calculate the diameter of $\tilde{\mathcal{F}}$. $\forall f \in \mathcal{F}$, given $(\boldsymbol{x}_i, y_i)$, let $\boldsymbol{x}_i^* = \inf_{\|\boldsymbol{x}_i - \boldsymbol{x}_i'\|_p \le \epsilon_p} y f(\boldsymbol{x}_i')$, and we let $\boldsymbol{x}_i^l$ be the output of $\boldsymbol{x}_i^*$ pass through the first to the $l-1$ layer, we have

$$
\begin{aligned}
|\tilde{f}(\boldsymbol{x}_i, y_i)| &= |\inf_{\|\boldsymbol{x}_i - \boldsymbol{x}_i'\|_p \le \epsilon_p} y f(\boldsymbol{x}_i')| \\
&= |W_d \rho(W_{d-1} \boldsymbol{x}_i^{d-1})| \\
&\overset{(i)}{\le} \|W_d\|_F \cdot \|\rho(W_{d-1} \boldsymbol{x}_i^{d-1})\|_2 \\
&= \|W_d\|_F \cdot \|\rho(W_{d-1} \boldsymbol{x}_i^{d-1}) - \rho(0)\|_2 \\
&\overset{(ii)}{\le} L_\rho M_d \|W_{d-1}(\boldsymbol{x}_i^{d-1})\|_2 \\
&\le \cdots \\
&\le L_\rho^{d-1} \prod_{l=2}^d M_l \|W_1 \boldsymbol{x}_i^*\|_2 \\
&\le L_\rho^{d-1} \prod_{l=1}^d M_l \cdot \|\boldsymbol{x}_i^*\|_2 \\
&\overset{(iii)}{\le} L_\rho^{d-1} \prod_{l=1}^d M_l \max\{1, q^{\frac{1}{2}-\frac{1}{p}}\}(\|\boldsymbol{X}\|_{p,\infty} + \epsilon),
\end{aligned}
$$

where inequality (i) is because of Lemma 3, inequality (ii) is because of the Lipschitz propertiy of activation function $\rho(\cdot)$, inequality (iii) is because of Lemma 2. Therefore, we have

$$
2 \max_{\tilde{f} \in \tilde{\mathcal{F}}} \|\tilde{f}\|_S = 2 \left( \frac{1}{m} \sum_{i=1}^m |\tilde{f}(\boldsymbol{x}_i, y_i)|^2 \right)^{\frac{1}{2}} \le 2 L_\rho^{d-1} \max\{1, q^{\frac{1}{2}-\frac{1}{p}}\}(\|\boldsymbol{X}\|_{p,\infty} + \epsilon) \prod_{l=1}^d M_l \overset{\triangle}{=} D.
$$

**Step 2: Distance to $\tilde{\mathcal{F}}^c$.** Let $\mathcal{C}_l$ be $\delta_l$-covers of $\{\|W_l\|_F \le M_l\}$, $l = 1, 2, \cdots, d$. Let

$$
\mathcal{F}^c = \{f^c : \boldsymbol{x} \to W_d^c \rho(W_{d-1}^c \rho(\cdots \rho(W_1^c \boldsymbol{x}) \cdots)), W_l^c \in \mathcal{C}_l, l = 1, 2 \cdots, d\}
$$

$$
\text{and } \tilde{\mathcal{F}}^c = \{\tilde{f} : (\boldsymbol{x}, y) \to \inf_{\|\boldsymbol{x} - \boldsymbol{x}'\|_p \le \epsilon} y f(\boldsymbol{x}') | f \in \mathcal{F}^c\}.
$$

For all $\tilde{f} \in \tilde{\mathcal{F}}$, we need to find the smallest distance to $\tilde{\mathcal{F}}^c$, i.e. we need to calculate the

$$
\max_{\tilde{f} \in \tilde{\mathcal{F}}} \min_{\tilde{f}^c \in \tilde{\mathcal{F}}^c} \|\tilde{f} - \tilde{f}^c\|_S.
$$

$\forall (\boldsymbol{x}_i, y_i), i = 1, \cdots, n$, given $\tilde{f}$ and $\tilde{f}^c$ with $\|W_l - W_l^c\|_F \le \delta_l$, $l = 1, \cdots, d$, consider

$$
\begin{aligned}
&|\tilde{f}(\boldsymbol{x}_i, y_i) - \tilde{f}^c(\boldsymbol{x}_i, y_i)| \\
=&|\inf_{\|\boldsymbol{x}_i - \boldsymbol{x}_i'\|_p} y_i f(\boldsymbol{x}_i') - \inf_{\|\boldsymbol{x}_i - \boldsymbol{x}_i'\|_p} y_i f^c(\boldsymbol{x}_i')|
\end{aligned}
$$

Let

$$
x_i^* = \arg \inf_{\|\boldsymbol{x}_i - \boldsymbol{x}_i'\|_p} y_i f(\boldsymbol{x}_i'), \quad \text{and} \quad x_i^c = \arg \inf_{\|\boldsymbol{x}_i - \boldsymbol{x}_i'\|_p} y_i f^c(\boldsymbol{x}_i'),
$$

we have

$$
\begin{aligned}
&|\tilde{f}(\boldsymbol{x}_i, y_i) - \tilde{f}^c(\boldsymbol{x}_i, y_i)| \\
=&|y_i f(\boldsymbol{x}_i^*) - y_i f^c(\boldsymbol{x}_i^c)| \\
=&|f(\boldsymbol{x}_i^*) - f^c(\boldsymbol{x}_i^c)|.
\end{aligned}
$$

Let

$$
z_i = \begin{cases} \boldsymbol{x}_i^c & if \quad f(\boldsymbol{x}_i^*) \ge f^c(\boldsymbol{x}_i^c) \\ \boldsymbol{x}_i & if \quad f(\boldsymbol{x}_i^*) < f^c(\boldsymbol{x}_i^c) \end{cases}
$$

Then,

$$
\begin{aligned}
&|\tilde{f}(\boldsymbol{x}_i, y_i) - \tilde{f}_c(\boldsymbol{x}_i, y_i)| \\
=&|f(\boldsymbol{x}_i^*) - f^c(\boldsymbol{x}_i^c)| \\
\leq&|f(z_i) - f^c(z_i)|.
\end{aligned}
$$

Define $g_b^a(\cdot)$ as

$$
g_b^a(z) = W_b\rho(W_{b-1}\rho(\cdots W_{a+1}\rho(W_a^c \cdots \rho(W_1^c z)\cdots))).
$$

In words, for the layers $b \geq l > a$ in $g_b^a(\cdot)$, the weight is $W_l$, for the layers $a \geq l \geq 1$ in $g_b^a(\cdot)$, the weight is $W_l^c$. Then we have $f(z_i) = g_d^0(z_i)$, $f(z_i) = g_d^L(z_i)$. We can decompose

$$
\begin{aligned}
&|f(z_i) - f^c(z_i)| \\
=&|g_d^0(z_i) - g_d^d(z_i)| \\
=&|g_d^0(z_i) - g_d^1(z_i) + \cdots + g_d^{d-1}(z_i) - g_d^d(z_i)| \\
\leq&|g_d^0(z_i) - g_d^1(z_i)| + \cdots + |g_d^{d-1}(z_i) - g_d^d(z_i)|.
\end{aligned}
\tag{11}
$$

To bound the gap $|f(z_i) - f^c(z_i)|$, we first calculate $|g_d^{l-1}(z_i) - g_d^l(z_i)|$ for $l = 1, \cdots, d$.

$$
\begin{aligned}
&|g_d^{l-1}(z_i) - g_d^l(z_i)| \\
=&|W_d\rho(g_{d-1}^{l-1}(z_i)) - W_d\rho(g_{d-1}^l(z_i))| \\
\overset{(i)}{\leq}&\|W_d\|_F\|\rho(g_{d-1}^{l-1}(z_i)) - \rho(g_{d-1}^l(z_i))\|_2 \\
\overset{(ii)}{\leq}&L_\rho M_d\|g_{d-1}^{l-1}(z_i) - g_{d-1}^l(z_i)\|_2 \\
\overset{(iii)}{=}&L_\rho M_d\|W_{d-1}\rho(g_{d-2}^{l-1}(z_i)) - W_{d-1}\rho(g_{d-2}^l(z_i))\|_2 \\
\leq&\cdots \\
\leq&L_\rho^{d-l}\prod_{j=l+1}^d M_j\|W_l\rho(g_{l-1}^{l-1}(z_i)) - W_l^c\rho(g_{l-1}^{l-1}(z_i))\|_2
\end{aligned}
$$

where (i) is due to Lemma 3, (ii) is due to the bound of $\|W_L\|$ and the Lipschitz of $\rho(\cdot)$, (iii) is because of the definition of $g_b^a(z)$. Then

$$
\begin{aligned}
&|g_d^{l-1}(z_i) - g_d^l(z_i)| \\
\leq&L_\rho^{d-l}\prod_{j=l+1}^d M_j\|W_l\rho(g_{l-1}^{l-1}(z_i)) - W_l^c\rho(g_{l-1}^{l-1}(z_i))\|_2 \\
=&L_\rho^{d-l}\prod_{j=l+1}^d M_j\|(W_l - W_l^c)\rho(g_{l-1}^{l-1}(z_i))\|_2 \\
\overset{(i)}{\leq}&L_\rho^{d-l}\prod_{j=l+1}^d M_j\|W_l - W_l^c\|_F\|\rho(g_{l-1}^{l-1}(z_i))\|_2 \\
\overset{(ii)}{\leq}&L_\rho^{d-l}\prod_{j=l+1}^d M_j\delta_l\|\rho(g_{l-1}^{l-1}(z_i))\|_2,
\end{aligned}
\tag{12}
$$

where inequality (i) is due to Lemma 3, inequality (ii) is due to Lemma 3 the assumption that $\|W_l - W_l^c\|_F \leq \delta_l$. It is lefted to bound $\|\rho(g_{l-1}^{l-1}(z_i)))\|_\infty$, we have

$$
\begin{aligned}
&\|\rho(g_{l-1}^{l-1}(z_i)))\|_2 \\
=&\rho(g_{l-1}^{l-1}(z_i))) - \rho(0)\|_2 \\
\leq& L_\rho \|g_{l-1}^{l-1}(z_i))\|_2 \\
=& L_\rho \|W_{l-1}^c \rho(g_{l-2}^{l-2}(z_i)))\|_2 \\
\leq& L_\rho \|W_{l-1}^c\|_F \|\rho(g_{l-2}^{l-2}(z_i)))\|_2 \\
\leq& L_\rho M_{l-1} \|\rho(g_{l-2}^{l-2}(z_i)))\|_2 \\
\leq& \cdots \\
\leq& L_\rho^{l-1} \prod_{j=1}^{l-1} M_j \max\{1, q^{\frac{1}{2}-\frac{1}{p}}\}(\|\boldsymbol{X}\|_{p,\infty} + \epsilon).
\end{aligned}
\tag{13}
$$

combining inequalities (12) and (13), we have

$$
\begin{aligned}
&|g_d^{l-1}(z_i) - g_d^l(z_i)| \\
\leq& L_\rho^{d-1} \frac{\prod_{j=1}^d M_j}{M_l} \delta_l \max\{1, q^{\frac{1}{2}-\frac{1}{p}}\}(\|\boldsymbol{X}\|_{p,\infty} + \epsilon) \\
=& \frac{D\delta_l}{2M_l}.
\end{aligned}
\tag{14}
$$

Therefore, combining inequalities (15) and (14), we have

$$
\begin{aligned}
&|f(z_i) - f^c(z_i)| \\
\leq& |g_d^0(z_i) - g_d^1(z_i)| + \cdots + |g_d^{d-1}(z_i) - g_d^d(z_i)| \\
\leq& \sum_{l=1}^d \frac{D\delta_l}{2M_l}.
\end{aligned}
\tag{15}
$$

Then

$$
\max_{\tilde{f} \in \tilde{\mathcal{F}}} \min_{\tilde{f}^c \in \tilde{\mathcal{F}}^c} \|\tilde{f} - \tilde{f}^c\|_S \leq \sum_{l=1}^d \frac{D\delta_l}{2M_l}.
$$

Let $\delta_l = 2M_l\varepsilon/dD, l = 1, \cdots, d$, we have

$$
\max_{\tilde{f} \in \tilde{\mathcal{F}}} \min_{\tilde{f}^c \in \tilde{\mathcal{F}}^c} \|\tilde{f} - \tilde{f}^c\|_S \leq \sum_{l=1}^d \frac{D\delta_l}{2M_l} \leq \varepsilon.
$$

**Step 3: Covering Number of $\tilde{\mathcal{F}}$.** We then calculate the $\varepsilon$-covering number $\mathcal{N}(\tilde{\mathcal{F}}, \|\cdot\|_S, \varepsilon)$. Because $\tilde{\mathcal{F}}^c$ is a $\varepsilon$-cover of $\tilde{\mathcal{F}}$. The cardinality of $\tilde{\mathcal{F}}^c$ is

$$
\mathcal{N}(\tilde{\mathcal{F}}, \|\cdot\|_S, \varepsilon) = |\tilde{\mathcal{F}}^c| = \prod_{l=1}^d |\mathcal{C}_l| \stackrel{(i)}{\leq} \prod_{l=1}^d \left(\frac{3M_l}{\delta_l}\right)^{h_l h_{l-1}} = \left(\frac{3dD}{2\varepsilon}\right)^{\sum_{l=1}^d h_l h_{l-1}},
$$

where inequality (i)) is due to Lemma 1.

**Step 4, Integration.** By Dudley's integral, we have

$$
\begin{aligned}
\mathcal{R}_S(\tilde{\mathcal{F}}) &\leq \frac{12}{\sqrt{m}} \int_0^{D/2} \sqrt{\log \mathcal{N}(\tilde{\mathcal{F}}, \|\cdot\|_S, \varepsilon)} d\varepsilon \\
&\leq \frac{12}{\sqrt{m}} \int_0^{D/2} \sqrt{\left(\sum_{l=1}^d h_l h_{l-1}\right) \log(3dD/2\varepsilon)} d\varepsilon \\
&= \frac{12D\sqrt{\sum_{l=1}^d h_l h_{l-1}}}{\sqrt{m}} \int_0^{1/2} \sqrt{\log(3d/2\varepsilon)} d\varepsilon.
\end{aligned}
\tag{16}
$$

Integration by part, we have

$$
\begin{aligned}
&\int_0^{1/2} \sqrt{\log(3d/2\varepsilon)}d\varepsilon \\
=&\frac{1}{2}\left(3\sqrt{\pi}\mathrm{erfc}(\sqrt{\log 3d}) + \sqrt{\log 3d}\right) \\
\leq&\frac{1}{2}\left(3\sqrt{\pi}\exp(-\sqrt{\log 3d}^2) + \sqrt{\log 3d}\right) \\
=&\frac{1}{2}\left(\frac{\sqrt{\pi}}{d} + \sqrt{\log 3d}\right) \\
\leq&\frac{1}{2}\left(2\sqrt{\log 3d}\right) \\
=&\sqrt{\log 3d}.
\end{aligned}
\tag{17}
$$

Plug equation (17) to equation (16), we have

$$
\mathcal{R}_S(\tilde{\mathcal{F}}) \leq \frac{24}{\sqrt{m}}\max\{1, q^{\frac{1}{2}-\frac{1}{p}}\}(\|\boldsymbol{X}\|_{p,\infty} + \epsilon)L_\rho^{d-1}\sqrt{\sum_{l=1}^d h_l h_{l-1}\log(3d)}\prod_{l=1}^d M_l.
$$

$\square$

## A.2    PROOF OF THEOREM 2

**Theorem 2** (($\|\cdot\|_{1,\infty}$-Norm Bound). *Given the function class $\mathcal{F}$ in equation (1) under $\|\cdot\|_{1,\infty}$-norm, and the corresponding adversarial function class $\tilde{\mathcal{F}}$ in equation (2). The adversarial Rademacher complexity of deep neural networks $\mathcal{R}_S(\tilde{\mathcal{F}})$ satisfies*

$$
\mathcal{R}_S(\tilde{\mathcal{F}}) \leq \frac{24}{\sqrt{m}}(\|\boldsymbol{X}\|_{p,\infty} + \epsilon)L_\rho^{d-1}\sqrt{\sum_{l=1}^d h_l h_{l-1}\log(3d)}\prod_{l=1}^d M_l.
$$

The proof is mimilar to the proof of the Frobenius norm bound. We first introduce the following inequality.

**Lemma 4.** *Let A be a $m \times n$ matrix and b be a $n$-dimension vector, we have*

$$
\|A \cdot b\|_\infty \leq \|A\|_{1,\infty}\|b\|_\infty.
$$

*Proof: let $A_i$ be the rows of A, $i = 1\cdots m$, we have*

$$
\|A \cdot b\|_\infty = \max|A_i b| \leq \max\|A_i\|_1\|b\|_\infty = \|A\|_{1,\infty}\|b\|_\infty.
$$

$\square$

**Step 1: Diameter of $\tilde{\mathcal{F}}$.**    We first calculate the diameter of $\tilde{\mathcal{F}}$. $\forall f \in \mathcal{F}$, given $(\boldsymbol{x}_i, y_i)$, let $\boldsymbol{x}_i^* = \inf_{\|\boldsymbol{x}_i - \boldsymbol{x}_i'\|_p \leq \epsilon_p} yf(\boldsymbol{x}_i')$, and we let $\boldsymbol{x}_i^l$ be the output of $\boldsymbol{x}_i^*$ pass through the first to the $l-1$ layer, we have

$$
\begin{aligned}
|\tilde{f}(\boldsymbol{x}_i, y_i)| &= |\inf_{\|\boldsymbol{x}_i - \boldsymbol{x}_i'\|_p \leq \epsilon_p} yf(\boldsymbol{x}_i')| \\
&= |W_d \rho(W_{d-1} \boldsymbol{x}_i^{d-1})| \\
&\overset{(i)}{\leq} \|W_d\|_{1,\infty} \cdot \|\rho(W_{d-1} \boldsymbol{x}_i^{d-1})\|_\infty \\
&= \|W_d\|_F \cdot \|\rho(W_{d-1} \boldsymbol{x}_i^{d-1}) - \rho(0)\|_\infty \\
&\overset{(ii)}{\leq} L_\rho M_d \|W_{d-1}(\boldsymbol{x}_i^{d-1})\|_\infty \\
&\leq \cdots \\
&\leq L_\rho^{d-1} \prod_{l=1}^{d} M_l \cdot \|\boldsymbol{x}_i^*\|_\infty \\
&\overset{(iii)}{\leq} L_\rho^{d-1} \prod_{l=1}^{d} M_l(\|\boldsymbol{X}\|_{p,\infty} + \epsilon),
\end{aligned}
$$

where inequality (i) is because of Lemma 4, inequality (ii) is because of the Lipschitz propertiy of activation function $\rho(\cdot)$, inequality (iii) is because of Lemma 2. Therefore, we have

$$
2 \max_{\tilde{f} \in \tilde{\mathcal{F}}} \|\tilde{f}\|_S = 2 \left( \frac{1}{m} \sum_{i=1}^{m} |\tilde{f}(\boldsymbol{x}_i, y_i)|^2 \right)^{\frac{1}{2}} \leq 2L_\rho^{d-1}(\|\boldsymbol{X}\|_{p,\infty} + \epsilon) \prod_{l=1}^{d} M_l \overset{\triangle}{=} D.
$$

**Step 2: Distance to $\tilde{\mathcal{F}}^c$.** Let $\mathcal{C}_l^i$ be $\delta_l$-covers of $\{\|W_l^i\|_1 \leq M_l\}$, $l = 1, 2, \cdots, d$, $i = 1, \cdots, h_l$, where $W_l^i$ is the $i^{th}$ row of $W_l^i$. Let

$$
\mathcal{F}^c = \{f^c : \boldsymbol{x} \to W_d^c \rho(W_{d-1}^c \rho(\cdots \rho(W_1^c \boldsymbol{x}) \cdots)), W_l^{ci} \in \mathcal{C}_l^i, i = 1, \cdots, h_l, l = 1, 2 \cdots, d\}
$$

$$
\text{and } \tilde{\mathcal{F}}^c = \{\tilde{f} : (\boldsymbol{x}, y) \to \inf_{\|\boldsymbol{x} - \boldsymbol{x}'\|_p \leq \epsilon} yf(\boldsymbol{x}') | f \in \mathcal{F}^c\}.
$$

For all $\tilde{f} \in \tilde{\mathcal{F}}$, we need to find the smallest distance to $\tilde{\mathcal{F}}^c$, i.e. we need to calculate the

$$
\max_{\tilde{f} \in \tilde{\mathcal{F}}} \min_{\tilde{f}^c \in \tilde{\mathcal{F}}^c} \|\tilde{f} - \tilde{f}^c\|_S.
$$

$\forall (\boldsymbol{x}_i, y_i), i = 1, \cdots, n$, given $\tilde{f}$ and $\tilde{f}^c$ with $|W_l^i - W_l^{ci}| \leq \delta_l, i = 1, \cdots, h_l, l = 1, \cdots, d$, we have $\|W_l - W_l^c\|_{1,\infty} \leq \delta_l$. By the same argument as the step 2 of the proof o Theorem 3, we have

$$
\max_{\tilde{f} \in \tilde{\mathcal{F}}} \min_{\tilde{f}^c \in \tilde{\mathcal{F}}^c} \|\tilde{f} - \tilde{f}^c\|_S \leq \sum_{l=1}^{d} \frac{D\delta_l}{2M_l}.
$$

Let $\delta_l = 2M_l \varepsilon / dD, l = 1, \cdots, d$, we have

$$
\max_{\tilde{f} \in \tilde{\mathcal{F}}} \min_{\tilde{f}^c \in \tilde{\mathcal{F}}^c} \|\tilde{f} - \tilde{f}^c\|_S \leq \sum_{l=1}^{d} \frac{D\delta_l}{2M_l} \leq \varepsilon.
$$

**Step 3: Covering Number of $\tilde{\mathcal{F}}$.** We then calculate the $\varepsilon$-covering number $\mathcal{N}(\tilde{\mathcal{F}}, \|\cdot\|_S, \varepsilon)$. Because $\tilde{\mathcal{F}}^c$ is a $\varepsilon$-cover of $\tilde{\mathcal{F}}$. The cardinality of $\tilde{\mathcal{F}}^c$ is

$$
\mathcal{N}(\tilde{\mathcal{F}}, \|\cdot\|_S, \varepsilon) = |\tilde{\mathcal{F}}^c| = \prod_{l=1}^{d} \prod_{i=1}^{h_l} |\mathcal{C}_l^i| \overset{(i)}{\leq} \prod_{l=1}^{d} \left( \frac{3M_l}{\delta_l} \right)^{h_l h_{l-1}} = \left( \frac{3dD}{2\varepsilon} \right)^{\sum_{l=1}^{d} h_l h_{l-1}},
$$

where inequality (i)) is due to Lemma 1.

**Step 4, Integration.** By the same argument as the step 4 of the proof o Theorem 1, integration by part, we have

$$\mathcal{R}_S(\tilde{\mathcal{F}}) \leq \frac{24}{\sqrt{m}}(\|\boldsymbol{X}\|_{p,\infty} + \epsilon)L_\rho^{d-1}\sqrt{\sum_{l=1}^{d} h_l h_{l-1} \log(3d)} \prod_{l=1}^{d} M_l.$$

$\square$

### A.3 PROOF OF THEOREM 3

**Theorem 3** (Lower Bound). *Given the function class $\mathcal{F}$ in equation (2), and the coresponding adversarial function class $\tilde{\mathcal{F}}$ in equation (1). Exist sample dataset S, s.t. the adversarial Rademacher complexity of deep neural networks $\mathcal{R}_S(\tilde{\mathcal{F}})$ satisfies*

$$\mathcal{R}_S(\tilde{\mathcal{F}}) \geq \Omega\left(\frac{(B+\epsilon)\prod_{l=1}^{d} M_l}{\sqrt{m}}\right).$$

The proof of the above Theorem is based on constructing a scalar network. By the definition of Rademacher complexity, if $\mathcal{H}'$ is a subset of $\mathcal{H}$, we have

$$\mathcal{R}_{\mathcal{S}}(\mathcal{H}') = \mathbb{E}_\sigma \frac{1}{m}\Big[\sup_{h \in \mathcal{H}'} \sum_{i=1}^{m} \sigma_i h(\boldsymbol{x}_i, y_i)\Big] \leq \mathbb{E}_\sigma \frac{1}{m}\Big[\sup_{h \in \mathcal{H}} \sum_{i=1}^{m} \sigma_i h(\boldsymbol{x}_i, y_i)\Big] = \mathcal{R}_{\mathcal{S}}(\mathcal{H}).$$

Therefore, it is enough to lower bound the complexity of $\tilde{\mathcal{F}}'$ in a particular distribution $\mathcal{D}$, where $\tilde{\mathcal{F}}'$ is a subset of $\tilde{\mathcal{F}}$. Let

$$\tilde{\mathcal{F}}' = \{\boldsymbol{x} \to \inf_{\|\boldsymbol{x}'-\boldsymbol{x}\|_p \leq \epsilon} y M_d \cdot M_2 w^T \boldsymbol{x} | w \in \mathbb{R}^q, \|w\|_2 \leq M_1\}.$$

We first prove that $\tilde{\mathcal{F}}'$ is a subset of $\tilde{\mathcal{F}}$. In $\tilde{\mathcal{F}}$, we let the activation function $\rho(\cdot)$ be a identity mapping. Let

$$W_1 = \begin{bmatrix} w \\ 0 \\ \vdots \\ 0 \end{bmatrix} \in \mathbb{R}^{h_1 \times h_0}, \quad W_l = \begin{bmatrix} M_l & 0 & \cdots & 0 \\ 0 & 0 & \cdots & 0 \\ \vdots & \vdots & & \vdots \\ 0 & 0 & \cdots & 0 \end{bmatrix} \in \mathbb{R}^{h_l \times h_{l-1}}, \quad l = 2, \cdots, d. \quad (18)$$

Then, we have $\|W_l\| \leq M_l$ and $\tilde{\mathcal{F}}$ with additional constraint in equation (18) reduce to $\tilde{\mathcal{F}}'$. In other words, $\tilde{\mathcal{F}}'$ is a subset of $\tilde{\mathcal{F}}$.

It turns out that we need to lower bound the adversarial Rademacher complexity of linear hypothesis. The results are given by the work of (Yin et al. (2019); Awasthi et al. (2020)). Below we state the result.

**Proposition 4.** *Given the function class $\mathcal{G} = \{\boldsymbol{x} \to y w^T \boldsymbol{x} | w \in \mathbb{R}^q, \|w\|_r \leq W\}$ and $\tilde{\mathcal{G}} = \{\boldsymbol{x} \to \inf_{\|\boldsymbol{x}'-\boldsymbol{x}\|_r \leq \epsilon} y w^T \boldsymbol{x} | w \in \mathbb{R}^q, \|w\|_r \leq W\}$, the adversarial Rademacher complexity $\mathcal{R}_S(\tilde{\mathcal{G}})$ satisfies*

$$\mathcal{R}_S(\tilde{\mathcal{G}}) \geq \max\left\{\mathcal{R}_S(\mathcal{G}), \frac{\epsilon \max\{1, q^{1-\frac{1}{r}-\frac{1}{p}}\}W}{2\sqrt{m}}\right\}.$$

Since the standard Rademacher complexity

$$\mathcal{R}_S(\mathcal{G}) = \frac{W}{m}\mathbb{E}_\sigma \|\sum_{i=1}^{m} \sigma_i \boldsymbol{x}_i\|_{r*},$$

let $\|\boldsymbol{x}_i\| = B$ with equal entries for $i = 1, \cdots, m$, by Lemma 2 we have

$$\mathcal{R}_S(\mathcal{G}) = \frac{W}{m}\mathbb{E}_\sigma |\sum_{i=1}^{m} \sigma_i| \max\{1, q^{1-\frac{1}{r}-\frac{1}{p}}\}B.$$

By Khintchine's inequality, we know that there exists a universal constant $c > 0$ such that

$$\mathbb{E}_\sigma |\sum_{i=1}^m \sigma_i| \geq c\sqrt{m}.$$

Then, we have

$$\mathcal{R}_S(\mathcal{G}) = \frac{cW}{\sqrt{m}} \max\{1, q^{1-\frac{1}{r}-\frac{1}{p}}\} B.$$

Therefore,

$$\mathcal{R}_S(\tilde{\mathcal{G}}) \geq \max\left\{\mathcal{R}_S(\mathcal{F}), \frac{\epsilon \max\{1, q^{1-\frac{1}{r}-\frac{1}{p}}\}W}{2\sqrt{m}}\right\}$$

$$\geq \frac{1}{1+2c}\mathcal{R}_S(\mathcal{F}) + \frac{2c}{1+2c} \times \frac{\epsilon \max\{1, q^{1-\frac{1}{r}-\frac{1}{p}}\}W}{2\sqrt{m}}$$

$$\geq \frac{c}{1+2c}\left(\frac{(B+\epsilon)\max\{1, q^{1-\frac{1}{r}-\frac{1}{p}}\}W}{\sqrt{m}}\right).$$

Let $W = \prod_{l=1}^d M_l$, we have

$$\mathcal{R}_S(\tilde{\mathcal{F}}) \geq \Omega\left(\frac{\max\{1, q^{1-\frac{1}{r}-\frac{1}{p}}\}(B+\epsilon)\prod_{l=1}^d M_l}{\sqrt{m}}\right),$$

where $r = 2$ for frobenius norm bound and $r = 1$ for $\|\cdot\|_{1,\infty}$-norm bound. $\qquad\square$

### A.4 PROOF OF THEOREM 4

**Theorem 4.** *Given the function class $\mathcal{F}$ in equation (2) under Frobbenius Norm, and the corresponding adversarial function class $\tilde{\mathcal{F}}$ in equation (1). The adversarial Rademacher complexity of deep neural networks $\mathcal{R}_S(\tilde{\ell}_{\mathcal{F}})$ satisfies*

$$\mathcal{R}_S(\tilde{\ell}_{\mathcal{F}}) \leq \frac{48K}{\gamma\sqrt{m}} \max\{1, q^{\frac{1}{2}-\frac{1}{p}}\}(\|\boldsymbol{X}\|_{p,\infty} + \epsilon)L_\rho^{d-1}\sqrt{\sum_{l=1}^d h_l h_{l-1} \log(3d)} \prod_{l=1}^d M_l.$$

The $(1, \infty)$-norm bound is similar, except the term $\max\{1, q^{\frac{1}{2}-\frac{1}{p}}\}$.

Proof: Firstly, we have

$$\tilde{\ell}(f(\boldsymbol{x}), y) = \max_{\|\boldsymbol{x}-\boldsymbol{x}'\|\leq\epsilon} \phi_\gamma(M(f(\boldsymbol{x}), y))$$

$$= \phi_\gamma(\inf_{\|\boldsymbol{x}-\boldsymbol{x}'\|\leq\epsilon} M(f(\boldsymbol{x}), y))$$

$$= \phi_\gamma(\inf_{\|\boldsymbol{x}-\boldsymbol{x}'\|\leq\epsilon} ([f(\boldsymbol{x})]_y - \max_{y'\neq y}[]f(\boldsymbol{x})]_{y'}))$$

$$= \phi_\gamma(\inf_{\|\boldsymbol{x}-\boldsymbol{x}'\|\leq\epsilon} \inf_{y'\neq y} ([f(\boldsymbol{x})]_y - [f(\boldsymbol{x})]_{y'}))$$

$$= \phi_\gamma(\inf_{y'\neq y} \inf_{\|\boldsymbol{x}-\boldsymbol{x}'\|\leq\epsilon} ([f(\boldsymbol{x})]_y - [f(\boldsymbol{x})]_{y'})).$$

$$= \max_{y'\neq y} \phi_\gamma(\inf_{\|\boldsymbol{x}-\boldsymbol{x}'\|\leq\epsilon} ([f(\boldsymbol{x})]_y - [f(\boldsymbol{x})]_{y'})).$$

Define

$$h^k(\boldsymbol{x}, y) = \inf_{\|\boldsymbol{x}-\boldsymbol{x}'\|\leq\epsilon} ([f(\boldsymbol{x})]_y - [f(\boldsymbol{x})]_k) + \gamma\mathbb{1}(y = k),$$

we now prove that

$$\max_{y'\neq y} \phi_\gamma(\inf_{\|\boldsymbol{x}-\boldsymbol{x}'\|\leq\epsilon} ([f(\boldsymbol{x})]_y - [f(\boldsymbol{x})]_{y'})) = \max_k \phi_\gamma(h^k(\boldsymbol{x}, y)).$$

If

$$\inf_{y'\neq y} \inf_{\|\boldsymbol{x}-\boldsymbol{x}'\|\leq\epsilon} ([f(\boldsymbol{x})]_y - [f(\boldsymbol{x})]_{y'}) \leq \gamma,$$

we have

$$\inf_{y' \neq y} \inf_{\|\boldsymbol{x} - \boldsymbol{x}'\| \leq \epsilon} ([f(\boldsymbol{x})]_y - f(\boldsymbol{x})]_{y'})) = \inf_k h^k(\boldsymbol{x}, y).$$

If

$$\inf_{y' \neq y} \inf_{\|\boldsymbol{x} - \boldsymbol{x}'\| \leq \epsilon} ([f(\boldsymbol{x})]_y - [f(\boldsymbol{x})]_{y'}) > \gamma,$$

we have

$$\phi_\gamma \big( \inf_{y' \neq y} \inf_{\|\boldsymbol{x} - \boldsymbol{x}'\| \leq \epsilon} ([f(\boldsymbol{x})]_y - [f(\boldsymbol{x})]_{y'})) = \phi_\gamma(\inf_k h^k(\boldsymbol{x}, y)) = 0.$$

Therefore, we have

$$\tilde{\ell}(f(\boldsymbol{x}), y) = \phi_\gamma(\inf_k h^k(\boldsymbol{x}, y)) = \max_k \phi_\gamma(h^k(\boldsymbol{x}, y)).$$

Define $\mathcal{H}^k = \{h^k(\boldsymbol{x}, y) = \inf_{\|\boldsymbol{x} - \boldsymbol{x}'\| \leq \epsilon}([f(\boldsymbol{x})]_y - f(\boldsymbol{x})]_k) + \gamma \mathbb{1}(y = k) | f \in \mathcal{F}\}$, we have

$$\mathcal{R}(\tilde{\ell}_\mathcal{F}) \overset{(i)}{\leq} K\mathcal{R}(\phi_\gamma \circ \mathcal{H}^k) \overset{(ii)}{\leq} \frac{K}{\gamma} \mathcal{R}(\mathcal{H}^k), \tag{19}$$

where inequality (i) is the Lemma 9.1 of (Mohri et al. (2018)), inequality (ii) is due to the Lipschitz property of $\phi_\gamma(\cdot)$. Now, define $f^k(\boldsymbol{x}, y) = \inf_{\|\boldsymbol{x} - \boldsymbol{x}'\| \leq \epsilon}([f(\boldsymbol{x})]_y - f(\boldsymbol{x})]_k)$, we have $h^k(\boldsymbol{x}, y) = f^k(\boldsymbol{x}, y) + \gamma \mathbb{1}(y = k)$. Define the function class

$$\mathcal{F}^k = \{f^k(\boldsymbol{x}, y) = \inf_{\|\boldsymbol{x} - \boldsymbol{x}'\| \leq \epsilon}([f(\boldsymbol{x})]_y - [f(\boldsymbol{x})]_k) | f \in \mathcal{F}\}.$$

We have

$$\begin{aligned}
\mathcal{R}(\mathcal{H}^k) &= \frac{1}{m} \mathbb{E}_\sigma \sup_{h^k \in \mathcal{H}^k} \sum_{i=1}^m \sigma_i h^k(\boldsymbol{x}_i, y_i) \\
&= \frac{1}{m} \mathbb{E}_\sigma \sup_{h^k \in \mathcal{H}^k} \sum_{i=1}^m \sigma_i \left[ f^k(\boldsymbol{x}, y) + \gamma \mathbb{1}(y = k) \right] \\
&= \frac{1}{m} \mathbb{E}_\sigma \sup_{h^k \in \mathcal{H}^k} \sum_{i=1}^m \sigma_i f^k(\boldsymbol{x}, y) + \frac{1}{m} \mathbb{E}_\sigma \sum_{i=1}^m \sigma_i \gamma \mathbb{1}(y = k) \\
&= \frac{1}{m} \mathbb{E}_\sigma \sup_{f^k \in \mathcal{F}^k} \sum_{i=1}^m \sigma_i f^k(\boldsymbol{x}, y) \\
&= \mathcal{R}(\mathcal{F}^k)
\end{aligned}$$

Finally, we need to bound the Rademacher complexity of $\mathcal{R}(\mathcal{F}^k)$. Notice that

$$[f(\boldsymbol{x})]_y - [f(\boldsymbol{x})]_k = (W_d^y - W_d^k)\rho(W_{d-1}(\rho(\cdots W_1(\boldsymbol{x}) \cdots))),$$

and we have $\|W_d^y - W_d^k\|_F \leq 2M_l$. By Theorem 3 (the results in binary classification case), we have

$$\mathcal{R}(\mathcal{F}^k) \leq \frac{48}{\sqrt{m}} \max\{1, q^{\frac{1}{2} - \frac{1}{p}}\}(\|\boldsymbol{X}\|_{p,\infty} + \epsilon)L_\rho^{d-1} \sqrt{\sum_{l=1}^d h_l h_{l-1} \log(3d)} \prod_{l=1}^d M_l. \tag{20}$$

Combining inequalities (20) and (19), we obtain that

$$\mathcal{R}_S(\tilde{\ell}_\mathcal{F}) \leq \frac{48K}{\gamma\sqrt{m}} \max\{1, q^{\frac{1}{2} - \frac{1}{p}}\}(\|\boldsymbol{X}\|_{p,\infty} + \epsilon)L_\rho^{d-1} \sqrt{\sum_{l=1}^d h_l h_{l-1} \log(3d)} \prod_{l=1}^d M_l.$$

$\square$

## B  DISCUSSION ON EXISTING METHODS FOR RADEMACHER COMPLEXITY

In this section, we discuss the related work, discuss the existing methods in calculating Rademacher complexity, and identify the difficulty of analyzing adversarial Rademacher complexity.

### B.1 Existing Methods

**'Layer Peeling' Bounds.** The main idea of calculating the Rademacher complexity of multi-layers neural networks is the 'peeling off' technique (Neyshabur et al. (2015)). We denote $g \circ \mathcal{F}$ as the function class $\{g \circ f | f \in \mathcal{F}\}$. By Talagrand's Lemma, we have $\mathcal{R}_S(g \circ f) \leq L_g \mathcal{R}_S(\mathcal{F})$. Based on this property, we can obtain $\mathcal{R}_S(\mathcal{F}_l) \leq 2L_\rho M_l \mathcal{R}_S(\mathcal{F}_{l-1})$, where $\mathcal{F}_l$ is the function class of $l$-layers neural networks. Since the Rdemacher complexity of linear function class is bounded by $\mathcal{O}(BM_1/\sqrt{m})$, we can get the upper bound $\mathcal{O}(B2^d L_\rho^{d-1} \prod_{l=1}^d M_l/\sqrt{m})$ by induction. We can remove the $L_\rho$ by assuming that $L_\rho = 1$ (e.g. Relu activation function).

Golowich et al. (2018) improves the dependence on depth-$d$ from $2^d$ to $\sqrt{d}$. The main idea is to rewrite the Rademacher complexity $\mathbb{E}_\sigma[\cdot]$ as $\mathbb{E}_\sigma \exp \ln[\cdot]$. Then, we can peel off the layer inside the $\ln(\cdot)$ function and the $2^d$ now appears inside the $\ln(\cdot)$.

**Covering Number Bounds.** Bartlett et al. (2017) uses a covering numbers argument to show that the generalization gap scale as

$$\tilde{\mathcal{O}}\left( \frac{B \prod_{l=1}^d \|W_l\|}{\sqrt{m}} \left( \sum_{l=1}^d \frac{\|W_l\|_{2,1}^{2/3}}{\|W_l\|^{2/3}} \right)^{3/2} \right),$$

where $\| \cdot \|$ is the spectral norm. The proof is based on the induction on layers. Let $W_l$ be the weight matrix of the present layer and $X_l$ be the output of $X$ pass through the first to the $l - 1$ layer. Then, one can compute the matrix covering number $\mathcal{N}(\{W_l X_l\}, \| \cdot \|_2, \epsilon)$ by induction.

**Adversarial Generalization Bounds.** Researchers have analyzed adversarial Rademacher complexity in linear and two-layers neural networks cases. In linear cases, the upper bounds can be directly derived by definition (Khim & Loh (2018); Yin et al. (2019)). In two-layers neural networks cases, an upper bound is derived using Massart's Lemma (Awasthi et al. (2020)). These proofs cannot be extended to multi-layers cases. Moreover, based on the definition of adversarial function class $\tilde{\mathcal{F}}$ in equation (1), the candidate functions are not composition functions, but with an $\inf$ operation in front of the neural networks. Then, the induction on layers seems not applicable in calculating adversarial Rademacher complexity for deep neural networks. The works of (Khim & Loh (2018)) and (Gao & Wang (2021)) indicate the difficulty of analyzing adversarial Rademacher complexity. They analyze other variants of adversarial Rademacher complexity of DNNs.

**'Tree Transformation' Bound.** Khim & Loh (2018) introduces a tree transformation $T$ and shows that $\max_{\|x-x'\| \leq \epsilon} \ell(f(\boldsymbol{x}), y) \leq \ell(Tf(\boldsymbol{x}), y)$. Then, we have the following upper bound for the adversarial population risk. For $\delta \in (0, 1)$,

$$\tilde{R}(f) \leq R(Tf) \leq R_m(Tf) + 2L\mathcal{R}_S(T \circ \mathcal{F}) + 3\sqrt{\frac{\log \frac{2}{\delta}}{2m}}.$$

It gives an upper bound of the robust population risk by the empirical risk and the standard Rademacher complexity of $T \circ f$. $\mathcal{R}_S(T \circ \mathcal{F})$ can be viewed as an approximation of adversarial Rademacher complexity. However, the empirical risk $R_m(Tf)$ is not the objective in practice. This analysis does not provide a guarantee for robust generalization gaps.

**FGSM Attack Bound.** The work of (Gao & Wang (2021)) tries to provide an upper bound for adversarial Rademacher complexity. To deal with the max operation in the adversarial loss, they consider FGSM adversarial examples. Then, the adversarial loss $\max_{\|\boldsymbol{x}'-\boldsymbol{x}\| \leq \epsilon} \ell(\boldsymbol{x}, y)$ becomes $\ell(f(\boldsymbol{x}_{FGSM}), y)$. By some assumptions on the gradient, they provide an upper bound for the Rademacher complexity of $\ell(f(\boldsymbol{x}_{FGSM}), y)$. However, the bound includes some parameters of the assumptions on the gradients, and FGSM underestimates the adversarial examples. It is hard to use this bound to analyze adversarial generalization. Therefore, the existing bounds give limited interpretations in understanding the generalization of adversarial training.

## B.2 WHY LAYER PEELING IS NOT APPLICABLE IN ADVERSARIAL SETTING?

We first take a look at the layer peeling technique.

$$
\begin{aligned}
\mathcal{R}_S(\mathcal{H}) &= \mathbb{E}_\sigma \frac{1}{m} \Big[ \sup_{h \in \mathcal{H}} \sum_{i=1}^m \sigma_i h(\boldsymbol{x}_i) \Big] \\
&= \mathbb{E}_\sigma \frac{1}{m} \Big[ \sup_{h' \in \mathcal{H}_{d-1}, \|W_d\| \le M_d} \sum_{i=1}^m \sigma_i W_d \rho(h'(\boldsymbol{x}_i)) \Big] \\
&\le M_d \mathbb{E}_\sigma \frac{1}{m} \Big[ \sup_{h' \in \mathcal{H}_{d-1}} \Big\| \sum_{i=1}^m \sigma_i \rho(h'(\boldsymbol{x}_i)) \Big\| \Big] \\
&\le 2 M_d L_\rho \mathbb{E}_\sigma \frac{1}{m} \Big[ \sup_{h' \in \mathcal{H}_{d-1}} \sum_{i=1}^m \sigma_i h'(\boldsymbol{x}_i) \Big] \\
&= 2 M_d L_\rho \mathcal{R}_S(\mathcal{H}_{d-1}),
\end{aligned}
$$

In adversarial settings, if we directly apply the layer peeling technique, we have

$$
\begin{aligned}
\mathcal{R}_S(\tilde{\mathcal{H}}) &= \mathbb{E}_\sigma \frac{1}{m} \Big[ \sup_{h \in \mathcal{H}} \sum_{i=1}^m \sigma_i \max_{\|\boldsymbol{x}_i - \boldsymbol{x}_i'\| \le \epsilon} h(\boldsymbol{x}_i') \Big] \\
&= \mathbb{E}_\sigma \frac{1}{m} \Big[ \sup_{h' \in \mathcal{H}_{d-1}, \|W_d\| \le M_d} \sum_{i=1}^m \sigma_i W_d \rho(h'(\boldsymbol{x}_i^*(h))) \Big] \\
&\le M_d \mathbb{E}_\sigma \frac{1}{m} \Big[ \sup_{h' \in \mathcal{H}_{d-1}} \Big\| \sum_{i=1}^m \sigma_i \rho(h'(\boldsymbol{x}_i^*(h))) \Big\| \Big] \\
&\le 2 M_d L_\rho \mathbb{E}_\sigma \frac{1}{m} \Big[ \sup_{h' \in \mathcal{H}_{d-1}} \sum_{i=1}^m \sigma_i h'(\boldsymbol{x}_i^*(h)) \Big] \\
&\ne 2 M_d L_\rho \mathbb{E}_\sigma \frac{1}{m} \Big[ \sup_{h' \in \mathcal{H}_{d-1}} \sum_{i=1}^m \sigma_i h'(\boldsymbol{x}_i^*(h')) \Big] \\
&= 2 M_d L_\rho \mathcal{R}_S(\tilde{\mathcal{H}}_{d-1}),
\end{aligned}
$$

where $\boldsymbol{x}_i^*(h)$ is the optimal adversarial example given a $d$-layers neural networks, $\boldsymbol{x}_i^*(h')$ is the optimal adversarial example given a $d-1$-layers neural networks. $\boldsymbol{x}_i^*(h) \ne \boldsymbol{x}_i^*(h')$ is the main reason why layer peeling cannot be directly extended to the adversarial settings.

This is the main reason why the work we introduce above studied the variants of adversarial Rademacher complexity. Once they take off the $\max$ operation by some approximation (e.g., let $\max_{\|x-x'\| \le \epsilon} \ell(f(\boldsymbol{x}), y) \le \ell(Tf(\boldsymbol{x}), y)$), they don't have the issue $\boldsymbol{x}_i^*(h) \ne \boldsymbol{x}_i^*(h')$ and they can use the layer peeling technique to bound the variants of adversarial Rademacher complexity. The main drop back is that they change the definition of adversarial Rademacher complexity. These bounds cannot provide theoretical guarantee on the robust generalization gap.

## B.3 WHY COVERING NUMBER CAN HELP AVOIDING THIS ISSUE?

In our opinion, it is hard to modify the procedure of layer peeling such that it is applicable in adversarial settings. Therefore, we try to bound the adversarial Rademacher complexity in a different way, using the covering number. In the proof of Theorem 1, we can see that we can avoid the issue of $\boldsymbol{x}_i^*(h) \ne \boldsymbol{x}_i^*(h')$. Specifically, when we calculate the covering number of the whole function class $\tilde{\mathcal{F}}$ directly, we only need to define the optimal adversarial examples $\boldsymbol{x}_i^*$ for a $d$-layer neural networks. We don't need to consider the optimal adversarial examples of neural networks with fewer layers. This is the benefit of covering numbers.

### B.4 COMPARISON OF DIFFERENT ADVERSARIAL GENERALIZATION BOUNDS

**VC-Dimension Bounds.** A classical approach in statistical learning is to use VC dimension to bound the generalization gap. It is thus natural to apply the VC-dim framework to adversarial setting, as (Cullina et al. (2018); Montasser et al. (2019); Attias et al. (2021)) did. However, these works did not provide a computable bound on the adversarial generalization gap, as explained next. let $\mathcal{H}$ be the hypothesis class (e.g. the set of neural networks with a given architecture).

In the work of (Cullina et al. (2018)), the authors defined adversarial VC-dim (AVC) and gave an bound on adversarial generalization gap with respect to $AVC(\mathcal{H})$. However, they did not show how to calculate AVC of neural works. Therefore, their paper did not provide a computable bound for adversarial generalization gap.

In the work of(Montasser et al. (2019)), the authors defined the adversarial function class as $\mathcal{L}_{\mathcal{H}}^{\mathcal{U}}$, where $\mathcal{L}$ is the loss and $\mathcal{U}$ is the uncertainty set. They bound the adversarial generalization gap by $\mathcal{L}_{\mathcal{H}}^{\mathcal{U}}$, which is different from $AVC(\mathcal{H})$ of (Cullina et al. (2018)). However, the authors did not provide a computable bound of as well, which means that their paper did not provide a computable bound of the adversarial adversarial generalization gap.

In the work of (Attias et al. (2021)), the authors assume that the perturbation set $U(\boldsymbol{x})$ is finite, i.e., for each sample $\boldsymbol{x}$, there are only $k$ adversarial examples that can be chosen. They showed that the adversarial generalization gap can be bounded by

$$\mathcal{O}\left(\frac{1}{\varepsilon^2}(\sqrt{kVC(\mathcal{H})}\log(\frac{3}{2}+a)kVC(\mathcal{H}))+\log\frac{1}{\delta}\right).$$

Note that there is a computable bound of $VC(\mathcal{H})$, which is the number of parameters, thus in terms of "computable", this bound is stronger than the previous two. However, this comes at a price: their bound depends on $k$, the number of allowed examined perturbed samples. This is a deviation from the original notion of adversarial generalization, where $U(\boldsymbol{x})$ is assumed to be an infinite set ($k \neq +\infty$). In contrast, our bound is for the "original" adversarial generalization gap, which allows $k = +\infty$.

**Adversarial Generalization Bounds in Other Settings.** The work of (Xing et al. (2021a;b); Javanmard et al. (2020)) study the generalization properties in the setting of linear regression. Gaussian mixture models are used to analyze adversarial generalization (Taheri et al. (2020); Javanmard et al. (2020); Dan et al. (2020)).

**Certified robustness.** A series of works study the certified robustness within the norm constraint around the original data. Cohen et al. (2019) privides an analysis on certified robustness via random smoothing. Lecuyer et al. (2019) studies certified robustness through the lens of differential privacy.

**Other Theoretical Studies on Adversarial Examples.** A series of works (Gilmer et al. (2018); Khoury & Hadfield-Menell (2018)) study the geometry of adversarial examples. The off-manifold assumption tells us that the adversarial examples leave the underlying data manifold (Szegedy et al. (2013)). Pixeldefends (Song et al. (2017)) uses a generative model to show that adversarial examples lie in a low probability region of the data distribution. The work of (Ma et al. (2018)) uses Local Intrinsic Dimensionality (LID) to argues that the adversarial subspaces are of low probability, and lie off the data submanifold.

## C ADDITIONAL EXPERIMENTS

In this section, we provide additional experiments.

### C.1 EXPERIMENTS ON VGG-11 AND VGG-13

In Figure 2, we show the experiments on VGG-11 and VGG-13. As we can see, the results are the same as the results in Figure 3, the gap of product of Frobenius norm between standard and adversarial training is large, which yields bad generalization.

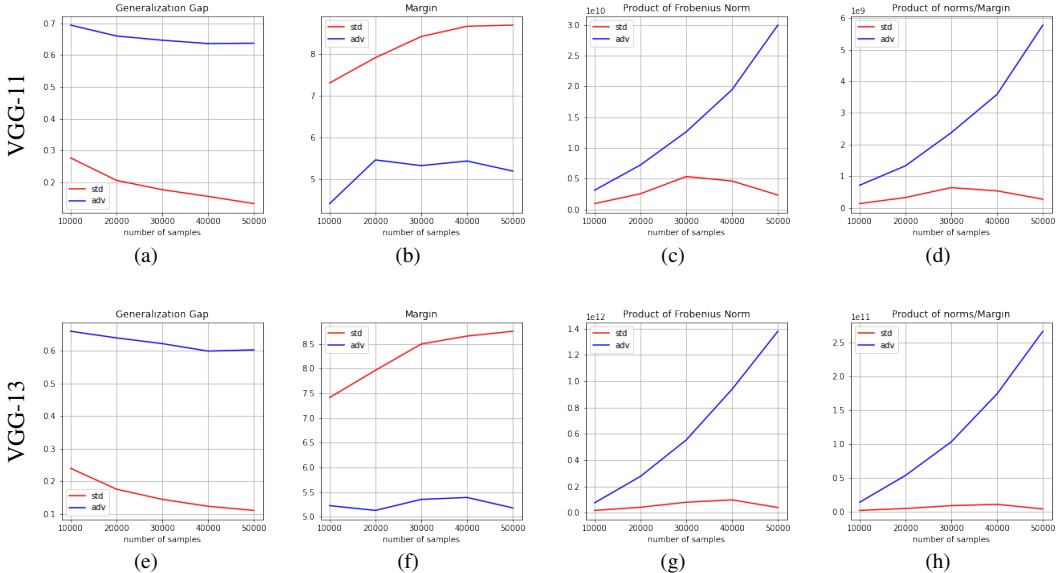

Figure 2: Product of the Frobenius norm in the experiments on VGG networks. The red lines are the results of standard training. The blue lines are the results of adversarial training. The first row are the experiments on VGG-11. The second row are the experiments on VGG-13. (a) and (e): Generalization gap. (b) and (f): Margin $\gamma$ over training set. (c) and (g): $\prod_{l=1}^{d} \|W_l\|_F$ of the neural networks. (d) and (h): $\prod_{l=1}^{d} \|W_l\|_F / \gamma$ of the neural networks.

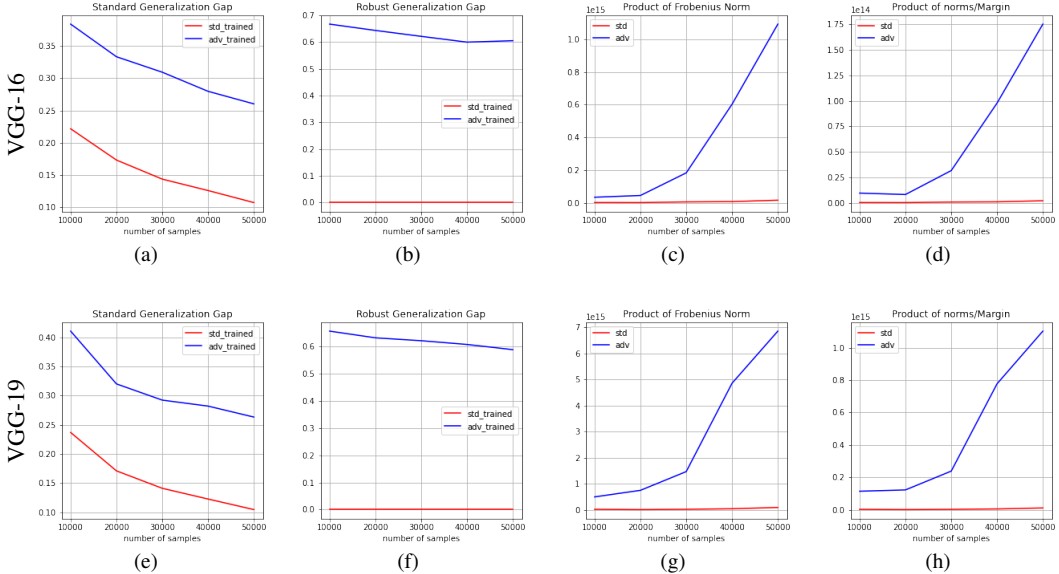

Figure 3: Product of the Frobenius norm in the experiments on CIFAR-10. The red lines are the results of standard training. The blue lines are the results of adversarial training. The first row is the experiments on VGG-16. The second row is the experiments on VGG-19. (a) and (e): Standard Generalization gap. (b) and (f): Robust Generalization Gap. (c) and (g): $\prod_{l=1}^{d} \|W_l\|_F$ of the neural networks. (d) and (h): $\prod_{l=1}^{d} \|W_l\|_F / \gamma$ of the neural networks.

$\|\cdot\|_{1,\infty}$**-Norm Bounds.** The $\|\cdot\|_{1,\infty}$-norm bounds are shown in Figure 4. Similar the the Frobenius norm bounds,The gap of $\prod_{l=1}^{d} \|W_l\|_{1,\infty}$ between adversarial training and standard training are large. But the magnitude of $\prod_{l=1}^{d} \|W_l\|_{1,\infty}$ is larger than the magnitude of $\prod_{l=1}^{d} \|W_l\|_F$.

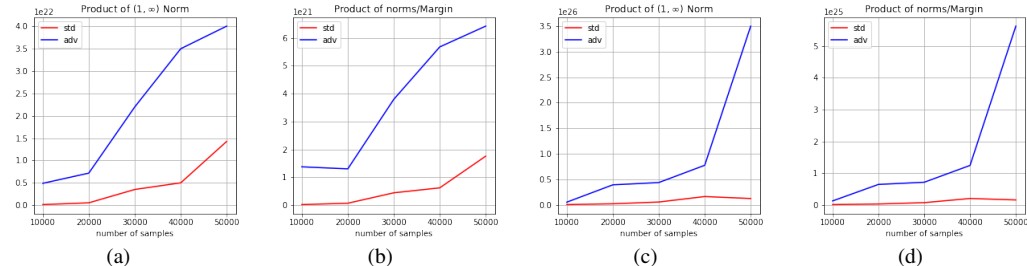

Figure 4: Product of the $\|\cdot\|_{1,\infty}$-Norm in the experiments on CIFAR-10. The red lines are the results of standard training. The blue lines are the results of adversarial training. (a) $\prod_{l=1}^{d}\|W_l\|_{1,\infty}$ of VGG-16 networks. (b) $\prod_{l=1}^{d}\|W_l\|_{1,\infty}/\gamma$ of VGG-16 networks. (c) $\prod_{l=1}^{d}\|W_l\|_{1,\infty}$ of VGG-19 networks. (d) $\prod_{l=1}^{d}\|W_l\|_{1,\infty}/\gamma$ of VGG-19 networks.

## C.2 ABLATION STUDY OF MARGINS

In Figure 5, we show the results of the margins in $1^{th}$, $3^{th}$, and, $5^{th}$-percentile of the training dataset. Since the (robust) training accuracy is 100%, the choice of percentile will not affect the results. As we can see in the Figure, in all the cases, the margins of standard training are larger than the margins of adversarial training. Since the margins appear in the divider in the upper bound of Rademacher complexity, the margins of the training dataset have some small effects on the bad generalization of adversarial training.

## C.3 EXPERIMENTS ON CIFAR-100

**Perfomance.**    In Table 2, we show the performance of standard training and adversarial training on CIFAR-100 using VGG-16 and 19 networks. We can see that using smaller number of training samples is unable to train an acceptable VGG-networks on CIFAR-100. Therefore it is hard use only 50000 training samples to study the trends of the weight norm using the experiments on CIFAR-100. We compare the product of weight norm between standard and adversarial training.

**Product of Weight Norms.**    In Figure 6, we show the results of on training VGG-19-16 and VGG-19 on CIFAR-100. Similar to the experiments on CIFAR-10, we can see that the adversarially trained models have larger weight norm that that of the standard trained model.

Table 2: Accuracy of standard and adversarial training on CIFAR-100 using VGG-16 and 19 networks. For standard training model, we shows the clean accuracy. For adversarial training model, we show the robust accuracy against PGD attacks.

| No. of Samples | 10000 | 20000 | 30000 | 40000 | 50000 |
|---|---|---|---|---|---|
| VGG-16-STD | 0.26 | 0.44 | 0.54 | 0.60 | 0.63 |
| VGG-16-ADV | 0.12 | 0.15 | 0.17 | 0.18 | 0.19 |
| VGG-19-STD | 0.32 | 0.47 | 0.53 | 0.58 | 0.62 |
| VGG-19-ADV | 0.12 | 0.16 | 0.17 | 0.19 | 0.21 |

## C.4 WEIGHT DECAY

The upper bounds of adversarial Rademacher complexity suggest adding a regularization term on the weights to improve generalization, which is essentially weight decay. In Figure 7, we provide the experiments of adversarial training with and without weight decay. In Figure 7 (a) and (c), we can see that adversarial training with weight decay has a smaller robust generalization gap. In Figure 7 (b) and (d), adversarial training with weight decay have a smaller product of weight norms. These experiments show the relationship between the robust generalization gap and the product of weight norms.

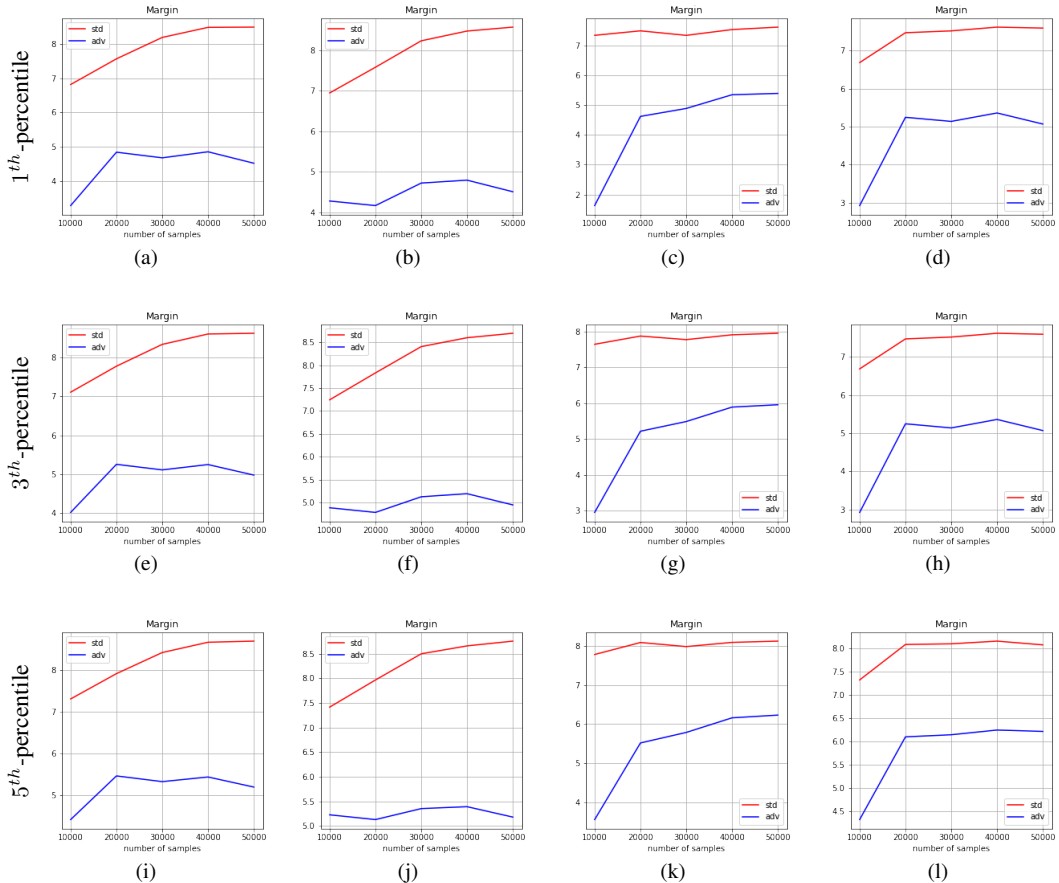

Figure 5: Ablation study of margins. The first to the $4^{th}$ rows are the experiments on VGG-11, 13, 16, and 19, respectively.

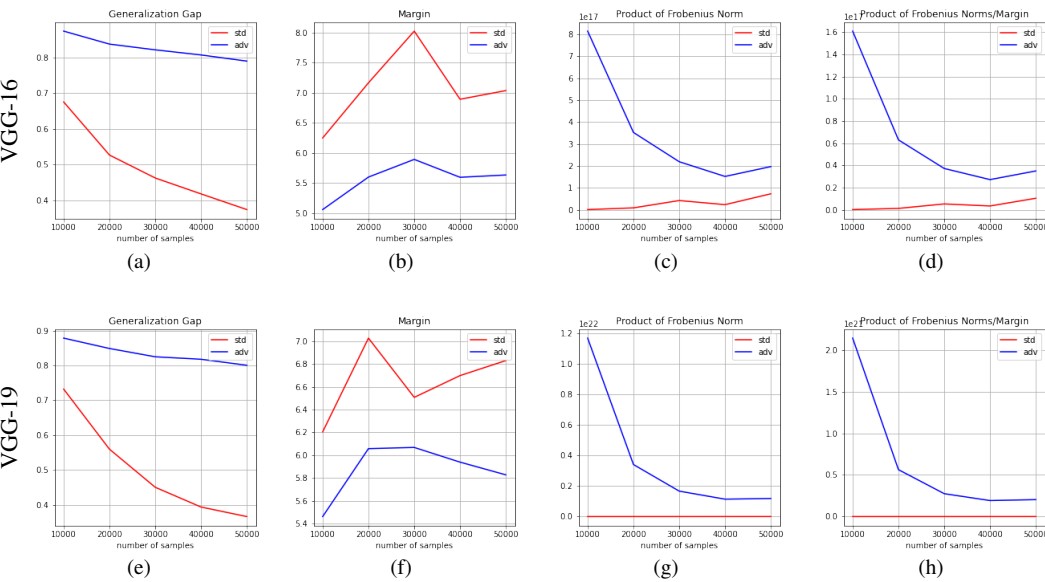

Figure 6: Product of the Frobenius norm in the experiments on VGG networks on CIFAR-100. The red lines are the results of standard training. The blue lines are the results of adversarial training. The first row are the experiments on VGG-16. The second row are the experiments on VGG-19. (a) and (e): Generalization gap. (b) and (f): Margin $\gamma$ over training set. (c) and (g): $\prod_{l=1}^{d} \|W_l\|_F$ of the neural networks. (d) and (h): $\prod_{l=1}^{d} \|W_l\|_F / \gamma$ of the neural networks.

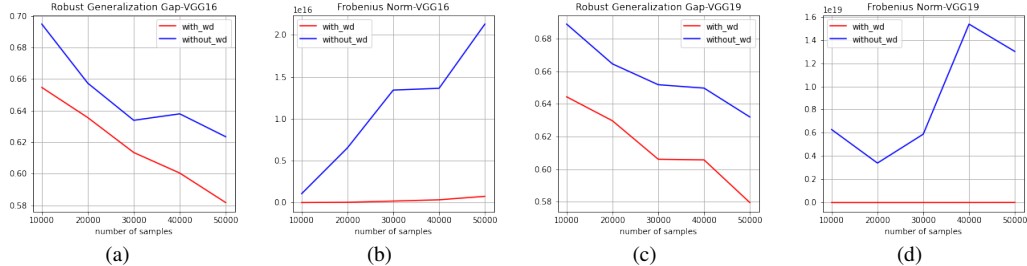

Figure 7: Experiments on the effects of weight decay. (a) Robust generalization gap with or without weight decay on VGG-16. (b) Frobenius norm with or without weight decay on VGG-16. (c) Robust generalization gap with or without weight decay on VGG-19. (d) Frobenius norm with or without weight decay on VGG-19.

## D    OPEN PROBLEM

In this section, we list some open problems.

**How to bridge the gap between the upper bound and the lower bound?**    There are two ways: one way is to show a depth/width-dependent lower bound as the reviewer suggested (increase lower bound); another way is to show a depth/width-independent upper bound (reduce upper bound). We briefly discuss which way is possible, and then discuss the technical challenges in both ways.

Which way is more likely to be true? If the upper bound can be improved to be depth-width-independent (thus matching our lower bound), then fundamentally the lower bound cannot be improved. Such a possibility exists. Actually, we are more inclined to this possibility, i.e., we tend to believe it is more promising to reduce the upper bound to be depth-width-independent, rather than increasing the lower bound. Anyhow, we don't have strong evidence of this possibility.

Technical challenge on increasing the lower bound (obtain a depth/width-dependent lower bound). In the current analysis, we construct a class of scalar networks to provide the lower bound. We obtain a closed-form expression of the adversarial examples. To obtain a depth/width-dependent lower bound, we need to: i) construct a more general function class of neural networks, and ii) then calculate the optimal adversarial examples in this class of neural networks. Currently, the challenge lies in the first step (construction). We have not tried hard to construct the function class so far, and we leave it to future work.

Can we reduce the upper bound (remove the dependence on depth/width in the upper bound)? This seems quite difficult by the current analysis. More specifically, the dependence $h\sqrt{dlogd}$ is probably unavoidable by our current approach of calculating the covering number. Despite the technical difficulty, we suspect that reducing the upper bound is doable by using a new tool other than covering number and layer peeling. This is surely nontrivial.

**Why adversarial training yields larger weight norms?**    In our opinion, it is because we require the additional capacity of the neural networks to fit the adversarial examples. As in the discussion of the work of (Neyshabur et al. (2017a)), we require more capacity of the model to fit random labels. A model with larger weight norms has a better ability to fit the training data. There might be other reasons, for example, the loss landscape of the minimax problem of adversarial training, the implicit bias of PGD attacks, or the implicit bias of adversarial training.

**Is the widely used regularization techniques essentially reducing weight norms?**    In adversarial training, there are many training tricks to reduce overfitting and yield better generalization, for example, stochastic weight averaging, early stopping, adversarial weight perturbation, and cyclic learning. It is an open problem that whether these techniques are related to the weight norms.

**How to design better algorithms to improve generalization?**    Our analysis suggests that adding regularization to the weight norms could improve generalization. The explicit regularization to

control the weight norms is weight decay, which is widely used. How to design implicit control on the weight norms is an open problem.

