# OpenReview forum: "Adversarial Rademacher Complexity of Deep Neural Networks"
_ICLR.cc/2022/Conference — ICLR 2022 Submitted_

### Official Review · Reviewer_Vmod · 2021-10-24

**Correctness:** 3
**Technical Novelty And Significance:** 3
**Empirical Novelty And Significance:** 2
**Recommendation:** 8
**Confidence:** 4

**Details Of Ethics Concerns:**

No concern

**Main Review:**

I think the main contribution of this paper is interesting. It directly overcomes the difficulties in deriving adversarial Rademacher complexity rather than using other variants. It provides both the upper bound and lower bound, both of which are important.

However, there are still a lot of things that can be improved in this paper. Below are my major concerns towards this paper (ordered in importance). My rating is currently weak accept given the importance of the upper and lower bounds, but it could be adjusted based on the author response towards my concerns.

[1] My understanding towards the adversarial training and the adversarial Rademacher complexity is that, both adversarial-trained and standard-trained neural networks have their own standard Rademacher complexity and adversarial Rademacher complexity. So "the generalization gap of adversarial training loss and adversarial testing loss" is larger than "the generalization gap of standard training loss and standard testing loss" for both adversarial-trained (denote as A and B) and standard-trained neural networks (denote as C and D), i.e. A>B and C>D. On the other hand, through experiments, it is observed that the adversarially trained neural networks obtains larger norms, so its generalization is worse, i.e. A>D. Is my understanding correct? Is it essential to provide evidence for A>C or B>D in other to conclude A>D? In Figure 1, what does the "generalization gap" refer to?

In addition, could you provide some insights on why the norm of adversarially trained model is larger? Why does increasing the samples size lead to a larger norm/margin? Also, given these observations, is there any way to improve the generalization performance of adversarial training? Is there any implications on the loss landscape of adversarial robust neural networks? The current Section 6 displays some observations but no detailed insights.

[1](as important as the above) The current proof of Theorem 3 only says "By the results of the lower bounds of ..., we obtain that ...". Please provide more details on the existing results and how to obtain the final result. Please provide some concrete illustrations to this either in pdf or in the discussion.

[3] The literature review of generalization of adversarial training in this paper only considers those about Rademacher complexity. There are many other studies working on the generalization from other theoretical aspects. Below are some important articles. Please do some literature research in this general area of theoretical study and include in this paper.

-----------------Papers which provide both upper bound and lower bound-----------------

Dan, Chen, Yuting Wei, and Pradeep Ravikumar. "Sharp statistical guaratees for adversarially robust gaussian classification." International Conference on Machine Learning. PMLR, 2020.

Xing, Yue, Ruizhi Zhang, and Guang Cheng. "Adversarially Robust Estimate and Risk Analysis in Linear Regression." International Conference on Artificial Intelligence and Statistics. PMLR, 2021.

-----------------Papers about generalization properties-----------------

Allen-Zhu, Zeyuan, and Yuanzhi Li. "Feature purification: How adversarial training performs robust deep learning." arXiv preprint arXiv:2005.10190 (2020).

Javanmard, Adel, Mahdi Soltanolkotabi, and Hamed Hassani. "Precise tradeoffs in adversarial training for linear regression." Conference on Learning Theory. PMLR, 2020.

Javanmard, Adel, and Mahdi Soltanolkotabi. "Precise statistical analysis of classification accuracies for adversarial training." arXiv preprint arXiv:2010.11213 (2020).

Taheri, Hossein, Ramtin Pedarsani, and Christos Thrampoulidis. "Asymptotic behavior of adversarial training in binary classification." arXiv preprint arXiv:2010.13275 (2020).

Wu, Dongxian, Shu-Tao Xia, and Yisen Wang. "Adversarial Weight Perturbation Helps Robust Generalization." Advances in Neural Information Processing Systems 33 (2020).

Xing, Yue, Qifan Song, and Guang Cheng. "On the Generalization Properties of Adversarial Training." International Conference on Artificial Intelligence and Statistics. PMLR, 2021.

Zhai, Runtian, et al. "Adversarially robust generalization just requires more unlabeled data." arXiv preprint arXiv:1906.00555 (2019).


[4] The main idea of this paper is not hard to follow, and the authors make a lot of comparison to existing literature. It would be great if the authors could make it clear about the following questions when describing the proof steps in Theorem 1:

   (1) Which steps are different from the derivation of standard Rademacher complexity? Which steps are not essential? If we use Theorem 1 and take \epsilon=0, what steps should we modify to obtain the standard Rademacher complexity mentioned in Section 5.3?

   (2) Which steps are different from the literature about adversarial Rademacher complexity? Which steps do they skip?

   (3) Could you explain the remark after before Theorem 2 in detail?



Minor issue (not ordered in importance):

[5] When mentioning your contributions in the last paragraph of Section 1, could you write some descriptions?
    (1) For the first contribution, is there any interesting findings in the bound?
    (2) For the second contribution, could you answer your why question?

[6] Please consider remove Proposition 1 and 2, and move Proposition 3 to the appendix. Proposition 1 and 2 do not help deepen the understanding of the main goal of this paper. Similarly, please shorten Section 3 for inequalities which are unrelated to the main goal.



**Summary Of The Paper:**

This paper overcomes some technical difficulties to provide adversarial Rademacher complexity of deep neural networks. Compared with existing literature which try to show other variants of adversarial Rademacher complexity, this paper directly works on adversarial Rademacher complexity itself. This paper provides both the lower bound and the upper bound. Besides, this paper conduct numerical experiments to combine with the theoretical bound to justify why adversarial training has a worse generalization than standard training.

**Summary Of The Review:**

This paper provides some important results about the adversarial Rademacher complexity so I vote for weak acceptance. But there are many issues towards the experiments, proofs, and the writing of this paper.

---

> ### Author Response · Authors · 2021-11-17
> **We thank the reviewer for the comments and suggestions. (1/2)**
>
> We thank the reviewer for the comments and suggestions. We appreciate the time you spent on the paper. Below we address the concerns and comments that you have provided.
>
> --updated 11/18, 11am Estern time, US--
>
> Q1: My understanding ... In Figure 1, what does the "generalization gap" refer to?
>
> A: This question is closely related to the first question of Reviewer DjdD. After more careful thought, we realize that our original discussion mixed a few concepts, and a better explanation is to perform an "ablation discussion".
>
> 1. Your understanding is correct. It is true that there are four kinds of generalization gaps as you described (with a slight difference that you mentioned "training loss and test loss", while people often use "training error and test error"). It is true that our experiments intend to discuss why "A>D".
>
> 2. In Figure 1, the red lines refer to the standard generalization gap of a standard-trained model (D). the blue lines refer to the robust generalization gap of an adversarially-trained model (A). In the current form, we only discuss A and D because it is more important in practice. Thanks for your suggestion, we think the discussion on B and C is also valuable since it provides a path for "ablation discussion". We add the discussion and plot all the four gaps in Figure 1 in the revised version.
>
> 3. As for the question "is it essential to provide evidence for A>C or B>D to conclude A>D", the simple answer is "Yes". We elaborate below.
>
> Following your notation, we denote four gaps:
>
> A: robust gen gap of an adv-trained model B: standard gen gap of an adv-trained model
> C: robust gen gap of a std-trained model   D: standard gen gap of a std-trained model.
>
> To explain the phenomenon A>D (by "ablation"), there are two possible paths: A
> ≥
>  B
> ≥
>  D and A
> ≥
>  C
> ≥
> D. We find that C is a degenerated case because the robust training error=100%. Since the model does not fit any adversarial examples in the training set, there is nothing to generalize to the adversarial examples in the test set. The generalization gap becomes meaningless. And the Rademacher complexity bound becomes a trivial bound. We will discuss the first path A
> ≥
>  B
> ≥
>  D.
>
> First, B>D is a widely observed phenomenon. It means adversarial training hurts standard generalization. This can be (at least partially) explained by the standard Rademacher complexity bounds. The weight norms in B (adversarially-trained) are larger than the weight norms in D (standard-trained), but not the constants (due to the same type of bound). Thus B>D can be partially explained by "adv-trained models have larger norms".
>
> Second, we empirically observe A>B. If we use the bounds of adv and std Rademacher complexity as an approximation of A and B, then A>B can be explained by the different constants of the two bounds, but not the weight norms (due to the same model).
>
> In summary, if we explain A>D by the Rademacher complexity bounds, then: (i) both the weight norm and the constants play a role; (ii) A>B can be explained by the different constants in our adversarial bound and the standard bound; (iii) B>D is explained by the different weight norms of the two models (standard-training v.s. adv-training), based on the standard bound.
>
> We can use simple formulas to explain. Robust gen-gap
> ≈
>  constant_adv * weight_norm. Standard gen-gap
> ≈
>  constant_std * weight _norm. To explain A > B > D, by the above two bounds, we only need to explain: constant_adv * weight_norm_adv > constant_std * weight_norm_adv > constant_std * weight_norm_std. The first is due to constant_adv > constant_std, and the second is due to weight_norm_adv > weight_norm_std.
>
> We can see that the weight norms are positively related to the robust generalization gap in both of the two paths. Thanks for the question again. We added the discussion on B and C in section 6 of the revised paper.
>
> Q2: In addition, could you ... but no detailed insights.
>
> A： Thanks for the question. We think these are all good and big questions for future works. We discuss the questions one by one.
>
> Q2.1: Why the norm of the adversarially trained model is larger?
>
> In our opinion, we think it is because we require additional capacity of the neural networks to fit the adversarial examples. As in the discussion of the work of [1], we require more capacity of the model to fit random labels. A model with larger weight norms has a better ability to fit the training data.
>
> Q2.2: Why does increasing the samples size lead to a larger norm/margin?
>
> The answer is similar to the last one. In our opinion, we require additional capacity of the neural networks to fit more adversarial examples.
>
> [1] Neyshabur B, Bhojanapalli S, McAllester D, et al. Exploring generalization in deep learning[J]. arXiv preprint arXiv:1706.08947, 2017.

---

> > ### Comment · Reviewer_Vmod · 2021-11-19
> > **Thanks for your response**
> >
> > The authors addressed all my concerns, therefore I raised my score.
> >
> > Some small typos:
> > Page 5: "Step 4, Integration." The other steps are in the form of "Step x: xxxx."
> > Page 28: "In this section, with list some open problems."

---

> > > ### Author Response · Authors · 2021-11-19
> > > **Thanks for the response.**
> > >
> > > Thanks for pointing out the typos, we will check the paper carefully and update a new version later.

---

> ### Author Response · Authors · 2021-11-17
> **We thank the reviewer for the comments and suggestions. (2/2)**
>
> Q2.3: Is there any way to improve the generalization performance of adversarial training?
>
> A: Based on the relationship between robust generalization gap and weight norms, we need a regularizer to control the weight norms. The explicit regularization of weight norms is weight decay. We provide a new set of experiments of adversarial training with and without weight decay in Appendix C.4. We show that adversarial training without weight decay has a larger generalization gap and smaller product of weight norms.
>
> As for the algorithm for implicit regularization, we think this is a good question and require further research. We list it as an open problem in Appendix D.
>
> Q2.4: Is there any implications on the loss landscape of adversarial robust neural networks?
>
> A: We don't know the answer now. It needs more experimental or theoretical verification. There might be other reasons, for example, the loss landscape of the minimax problem of adversarial training, the implicit bias of PGD attacks, or the implicit bias of adversarial training. Thanks for the question. We discuss it as an open problem in Appendix D.
>
> Q3: The current proof of Theorem 3 ... either in pdf or in the discussion.
>
> A: Thanks for the question. We cite the results of the reference paper as proposition 4. We provide a more detailed proof of the lower bound in Appendix A.3 in the revised paper.
>
> Q4: The literature review of generalization of adversarial training in this paper only considers those about Rademacher complexity.
>
> A: Thanks for the suggenstion. In the current form, we discussion the adversarial generalization other than Rademacher complexity in Appendix. In the revised paper, we move it to the main paper. We provide the discussion of the paper you mentioned in Section 3.
>
> Q5: The main idea of this paper is not hard to follow ... in Theorem 1:
>
> (1) Which steps are different from ... in Section 5.3?
>
> A: Most steps are different. The proof of standard Rademacher complexity is based on layer peeling. Our proof of adversarial Rademacher complexity is based on the covering number. The four steps summarize how to calculate the covering number, and all four steps are not related to the layer peeling. We cannot get the layer peeling bounds by modifying the calculation of the covering number. If we take \epsilon=0, we get a loose bound for SRC.
>
> (2) Which steps are different from the literature about adversarial Rademacher complexity? Which steps do they skip?
>
> A: It is not about skipping any steps. In the literature about adversarial Rademacher complexity, they consider variants of ARC such that they can use layer peeling. In our work, we consider other methods since we cannot use layer peeling.
>
> (3) Could you explain the remark before Theorem 2 in detail?
>
> A: The remark tries to explain why layer peeling seems not applicable in adversarial settings. Why using covering numbers (steps 1-4) can overcome this issue? The reason can be found in step 2. It is hard to describe the details without mathematics. We provide more details in Appendix B.2 and B.3 in the revised paper.
>
> In Appendix B.2, we carefully discuss the difficulty of applying layer peeling in adversarial settings.
>
> In Appendix B.3, we discuss why using covering numbers can overcome this issue.
>
> Minor issue:
>
> 1, When mentioning ... could you answer your why question?
>
> A: Thanks for the suggestion. We list our contributions in more detail.
>
> 3, Please consider remove ... the main goal.
>
> A: Thanks for the suggestion. The detailed descriptions in Section 3 are moved to Appendix B.1. As for Proposition 1 and 2 and  Proposition 3, we think it is important. Without them, the readers don't know why we need to consider Rademacher complexity and why we need to consider covering numbers. Based on these considerations, we keep it in the main paper.

---

### Official Review · Reviewer_dFnk · 2021-11-02

**Correctness:** 4
**Technical Novelty And Significance:** 3
**Empirical Novelty And Significance:** 2
**Recommendation:** 8
**Confidence:** 3

**Main Review:**

Strengths:
- Relevant topic of theoretical investigation of adversarial examples and adversarial training, that has gained traction in recent years.
- Improved theoretical result over prior art.
- Meaningful experiments based on the theoretical results, that suggest some reasons why adversarial training does not generalize well.
- Clear and well-written paper (a few typos remain).

Weaknesses:
- It would be great if the paper included a comparison to existing adversarial bounds based on Rademacher complexity or other frameworks, e.g., plotted for a toy example or small network. Lacking this, it is harder to judge the improvement the current paper makes over prior results in terms of tightness of the bound. Moreover, it might be worth comparing the contribution to other types of theoretical approaches in the field, e.g., the provable methods that the paper cites.

Questions and other comments:
- It would be good to underline that the bounds provided also hold for convolutional neural networks earlier than the experiments section (or, more generally, what layers are covered).
- What is the impact of using PGD adversarial examples in practice instead of the optimal perturbation?
- Is it reasonable to consider a loss in the range 0-1 for neural networks?
- Are Rademacher complexity bounds tight enough for neural networks to be informative or applicable in practice?

[Update post discussion] I am raising my rating by one point, following the exchanges below.

**Summary Of The Paper:**

This paper proposes new generalization bounds for adversarial training in neural networks based on the Rademacher complexity. These are more general than previous results in that they apply to neural networks of any depth. Experiments are performed on CIFAR-10 and CIFAR-100 with multiple VGG architectures. These, related with the main quantities appearing in the proposed bounds, provide an explanation for the limited generalization capacity of adversarial training.

**Summary Of The Review:**

Good theoretical result supported by experimental evaluation on relevant topic.

---

> ### Author Response · Authors · 2021-11-17
> **We thank the reviewer for the comments and suggestions. (1/2)**
>
> We thank the reviewer for the comments and suggestions. We appreciate the time you spent on the paper. Below we address the concerns and comments that you have provided.
>
> --updated 11/19, 7am Estern time, US--
>
> ---
> **Q1**: It would be great if the paper included a comparison to existing adversarial bounds based on Rademacher complexity or other frameworks, e.g., plotted for a toy example or small network. Lacking this, it is harder to judge the improvement the current paper makes over prior results in terms of tightness of the bound. Moreover, it might be worth comparing the contribution to other types of theoretical approaches in the field, e.g., the provable methods that the paper cites.
>
> A: Thanks for the comment. To our knowledge, our work provides the first computable bound for the "original" adv-gen-gap (adversarial generalization gap) in multi-layer networks. Our contribution is not to provide "improvement in terms of tightness of the bound", but to provide the first such bound. There are some other bounds for adversarial generalization, but they are not the bounds on the "original" adv-gen-gap, thus are not comparable to ours. This point was mentioned in Sec. 3 and App. B, but perhaps not clear enough. Let us explain below.
>
> To provide a big picture, we can classify existing works into three types.
>
>    **First type of works**: Radamacher complexity bounds on simplified models, such as linear models, or two-layer network. Compared to these works, as the reviewer mentioned in the summary, we provide the first such bound for multi-layer network (using new techniques).
>
>   **Second type of works**: Variants of adversarial Radamacher complexity bounds. Khim & Loh (2018) and Gao & Wang (2021) studied some variants, but both bounds do not lead to a bound on the original version of adversarial generaization gap. As mentioned in App. B.1.,  Khim & Loh (2018) provided a bound on the difference between generalization error and a "tree-transformed training error", which is not the original training error, thus it is not clear whether their bound is an upper bound of the adv-gen-gap (which shall be the difference between generalization error and original training error). Gao & Wang (2021) provided a generalization bound between the error on FGSM-attacked-samples (not the original generalization error) and the training error, thus their bound is also not an upper bound of the adv-gen-gap.
>
>    **Third type of works**: VC dimension. A classical approach in statistical learning is to use VC dimension to bound the generalization gap. It is thus natural to apply the VC-dim-framework to adversarial setting, as [1][2][3] did. However, these works did not provide a computable bound on the adversarial gen-gap, as explained next. let H be the hypothesis class (e.g. the set of neural networks with a given architecture).
>
> In [1], the authors defined adversarial VC-dim (AVC) and gave an bound on adversarial gen-gap with respect to AVC(H). However, they did not show how to calculate AVC of neural works. Therefore, their paper did not provide a **computable** bound for adversarial gen-gap.
>
> In [2], the authors defined the adversarial function class as $L_H^U$, where L is the loss and U is the uncertainty set.  They bound the adversarial gen-gap by $VC(L_H^U)$, which is different from AVC(H) of [1]. However, the authors did not provide a computable bound of $VC(L_H^U)$ as well, which means that their paper did not provide a computable bound of the adversarial gen-gap.
>
> In [3] the authors assume that the perturbation set U(x) is finite, i.e., for each sample x, there are only k adversarial examples that can be chosen. They showed that the adversarial gen-gap can be bounded by poly(k, VC(H)). Note that VC(H) is different from VC(L_H^U) in [2] and AVC(H) in [1]. There is a computable bound of VC(H), which is the number of parameters, thus in terms of "computable", this bound is stronger than [1] and [2]. However, this comes at a price: their bound depends on k, the number of allowed examined perturbed samples. This is a deviation from the original notion of adversarial generalization, where U(x) is assumed to be an infinite set ($k=+\infty$). In contrast, our bound is for the "original" adversarial generaization gap, which allows $k = \infty$.

---

> > ### Comment · Reviewer_dFnk · 2021-11-23
> > **Thank you for the detailed explanations**
> >
> > I would like to thank the authors for taking the reviewers' comments on board, but also for their thoughtful answers. Most of my questions were open ended, and I deem have been answered in detail. I would like to raise my score by one point.
> >
> > Regarding Q1, I think it makes sense to compare the proposed bounds with existing Rademacher bounds for simplified models (not for the current manuscript, more of a discussion on principle). Of course, the setup would have to be limited to what these prior bounds are able to address, and with the side note that the results in the paper are applicable to a broader class of networks.

---

> > > ### Author Response · Authors · 2021-11-24
> > > **Thank you for the positive feedback.**
> > >
> > > Thank you for the positive feedback. We really appreciate it.
> > >
> > > Following your suggestion, we will add a comparison to the bounds for linear and 2-layers neural nets to the paper; we also attach the comparison below for your convenience. We understand that the comparison would help readers better understand the relation/difference of our bound and earlier works (more specifically, the first type of works), thus will be a useful plus to the paper. Thank you for the valuable suggestion.
> > >
> > > 1) Comparison to the bounds for linear models in [1].
> > >
> > > For a linear model (can be viewed as 1-layer neural nets), our bound reduces to $O((B+\epsilon)hW/\sqrt{m})$, and the bound in [1] is $O((B+\epsilon)W/\sqrt{m})$. The bound in [1] is tighter by a factor of $h$ since their bound is directly derived from the definition of adversarial Rademacher complexity.
> > >
> > > 2. Comparison to the bounds for the 2-layers case in [2].
> > >
> > > In the case of 2-layer neural networks, our bound becomes $O((B+\epsilon)h M_1M_2/ \sqrt{m})$, and the bound in [2] is $O((B+\epsilon)h M_1M_2 \sqrt{\log m/ m})$. In this case, our bound is tighter: we reduce the dependence on the sample size from $\sqrt{\log m/ m}$ to $\sqrt{1/ m}$. The difference is due to the different proof techniques: [2] used Massart's Lemma, while we use the covering number. Note that both bounds have a dependence on $h$.
> > >
> > > [1] Yin D, Kannan R, Bartlett P. Rademacher complexity for adversarially robust generalization[C]//International conference on machine learning. PMLR, 2019: 7085-7094.
> > >
> > > [2] Awasthi P, Frank N, Mohri M. Adversarial learning guarantees for linear hypotheses and neural networks[C]//International Conference on Machine Learning. PMLR, 2020: 431-441.

---

> ### Author Response · Authors · 2021-11-19
> **We thank the reviewer for the comments and suggestions. (2/2)**
>
>  (continue) To summarize, we are aware of three papers in the area that have provided a certain computable bound for multi-layer neural-net: Khim & Loh (2018), Gao & Wang (2021)) and [3]. However, none of them provided a bound on the original adv-gen-gap. (i) The bound of Khim & Loh (2018) is not a bound on the difference of adversarial training error and adversarial generalization error, i.e., not a bound on the adv-gen-gap; (ii) The bounds of Gao & Wang (2021)) and [3] are not for the original adv-gen-gap (defined for infinitely many perturbed samples) but finitely many perturbed samples. These computable bounds and our bound can be calculated (to get concrete numbers and plotted), but since those bounds do not form an upper bound of the adv-gen-gap, it would be an unfair comparison. That being said, if the reviewer is aware of a work on the original adv-gen-gap, we would be happy to add and compare.
>
> In short, to our knowledge, the main contribution of our work is to provide the first computable bound on the original adv-gen-gap for multi-layer network. Hope this discussion has clarified this point. Thanks again for the great comment.
>
> [1] Cullina D, Bhagoji A N, Mittal P. PAC-learning in the presence of evasion adversaries[J]. arXiv preprint arXiv:1806.01471, 2018.
>
> [2] Montasser O, Hanneke S, Srebro N. Vc classes are adversarially robustly learnable, but only improperly[C]//Conference on Learning Theory. PMLR, 2019: 2512-2530.
>
> [3] Attias I, Kontorovich A, Mansour Y. Improved Generalization Bounds for Adversarially Robust Learning[J]. 2021.
>
> ---
> **Q2**: It would be good to underline that the bounds provided also hold for convolutional neural networks earlier than the experiments section (or, more generally, what layers are covered).
>
> A: Thanks for the suggestion. In the revised paper, we underlined it in Section 2 when we discussed the hypothesis class.
>
> ---
> **Q3**: What is the impact of using PGD adversarial examples in practice instead of the optimal perturbation?
>
> A: Thanks for the question.
>
>   i) Using PGD adversarial examples will affect the estimation of robust training error and test error, and then affect the estimation of robust generalization gap. To be specific, when we calculate the training error on PGD adversarial examples other than optimal adversarial examples, the training error is underestimated. Similarly, the test error is also underestimated. We can view the gap computed based on PGD adversarial examples as an approximation of the adversarial generalization gap (defined by the optimal adversarial examples). Nevertheless, we don't know how good this approximation is.
>
>   ii) It is not easy to compute the optimal perturbation for now, thus for practical purpose we need to pick one type of attacks to aprpoximate the optimal attack. There are multiple types of attacks in the area; due to limited time, and also due to popularity of PGD attacks, we only use PGD attacks.
>
>   iii) That being said, there may be other methods that can better approximate the optimal attack (either in the area now, or in the future). It is possible to use these methods to compute a better approximation of the generalization gap in practice. We can still use our framework to conduct the comparison: in the experiment section, we only need to change the first two figures on the empirical robust generalization gap, and the rest of the section is similar.
>
> ---
> **Q4**: Is it reasonable to consider a loss in the range 0-1 for neural networks?
>
> A: We use the range in 0-1 to simplify the notation. Propositions 1 and 2 also hold for a loss in the range [0,C] for C>0 with a small change: there will be an additional multiplier C in the second and the third terms on the right-hand side of the bound. We modify Propositions 1 and 2 in the revised paper in Section 2.
>
> ---
> **Q5**: Are Rademacher complexity bounds tight enough for neural networks to be informative or applicable in practice?
>
> A: We suppose the reviewer is asking about Rademacher complexity bounds for standard training of neural networks (correct us if not). This is a very important question in learning theory. We separate the two notions "tight" and "informative". Some researchers point out that Rademacher complexity bounds are not tight (e.g., [5]). Nevertheless, some works show that Rademacher complexity bounds can provide some meaningful bounds that can explain practical training, e.g., [4].
>
> [4] Neyshabur B, Bhojanapalli S, McAllester D, et al. Exploring generalization in deep learning[J]. arXiv preprint arXiv:1706.08947, 2017.
> [5] Zhang C, Bengio S, Hardt M, et al. Understanding deep learning (still) requires rethinking generalization[J]. Communications of the ACM, 2021, 64(3): 107-115.

---

### Official Review · Reviewer_DjdD · 2021-11-03

**Correctness:** 3
**Technical Novelty And Significance:** 3
**Empirical Novelty And Significance:** 2
**Recommendation:** 6
**Confidence:** 3

**Main Review:**

Strengths:
1.	This paper provides new bounds for adversarial Rademacher complexity.
2.	This paper is well written and easy to read.

Weaknesses:

The empirical validation in this paper seems insufficient:

1. Although the authors verified that the product of weight norm in adversarial training is indeed much larger than that in standard training, I think this is not a strong empirical verification of the proposed bound. The reason is there existing some constants in the bounds in adversarial training and standard training, which may be different. Therefore, simply comparing the product of the weight norm is not rigorous. I would like to see more discussion on this. We know that some of the existing deep learning theory may be only mathematically correct, thus providing sufficient empirical evidence to show the consistence between the theory and practice is important.

2. There exists a product of the weight norm in the proposed bound, which implies that the trained model can generalize better if this product is smaller. Therefore, can we use some techniques to regularize this product during training to improve the generalization ability. The authors are recommended to give such kind of experiments to support their theoretical results.

====after rebuttal===

After reading the response from the authors, I raised my rating.

**Summary Of The Paper:**

In this paper, the authors developed an upper bound of adversarial Rademacher complexity, which includes the product of weight norms. This implies that the large weight norm hinders from achieving good generalization performance in adversarial. The authors also empirically show that the product of weight norm in adversarial training is indeed much larger than that in standard training.

**Summary Of The Review:**

1. This paper provides a new complexity bound for adversarial training.
2. The empirical validation of this paper is insufficient.

---

> ### Author Response · Authors · 2021-11-17
> **We thank the reviewer for the comments and suggestions.**
>
> --updated 11/17, 10pm Estern time, US--
>
> We thank the reviewer for the comments and suggestions. We appreciate the time you spent on the paper. Below we provide a response to the concerns and comments that you have provided.
>
> Q1: Although the authors ... I would like to see more discussion on this.
>
> A: Thanks for raising an excellent point. This question is closely related to the first question of Reviewer Vmod. After more careful thought, we realize that our original discussion mixed a few concepts, and a better explanation is to perform an "ablation discussion".
> More specifically, we will use the four notions of "gaps" mentioned by Reviewer Vmod.
>
> Standard generalization gap (standard gen-gap): the generalization gap of standard training loss and standard testing loss.
>
> Robust generalization gap (robust gen-gap): the generalization gap of adversarial training loss and adversarial testing loss.
>
>  A: robust gen-gap for adversarial-trained net      |  B: standard gen-gap for adversarial-trained net
>  C: robust gen-gap for standard-trained net         |  D: standard gen-gap for standard-trained net
>
> Our original paper did not explicitly summarize these four notions. Readers may get an impression that what we were claiming is "we empirically observe A>D; based on our bound, an explanation is that the product of weight norm is larger in adv-trained net". The reviewer raised a valid concern on this claim. This claim is over-simplified since A and D differ in two aspects (different algorithms, and different types of gen-gaps). Their comparison shall be decomposed in two steps.
>
> To explain the phenomenon A>D (by "ablation"), there are two possible paths: A $\ge$ B $\ge$ D and A$\ge$ C $\geq$D. We will discuss the first path A $\ge$ B $\ge$ D.
>
> First, B>D is a widely observed phenomenon. It means adversarial training hurts standard generalization. This can be (at least partially) explained by the standard Rademacher complexity bounds. The weight norms in B (adversarially-trained) are larger than the weight norms in D (standard-trained), but not the constants (due to the same type of bound). Thus B>D can be partially explained by "adv-trained models have larger norms".
>
> Second, we empirically observe A>B. If we use the bounds of adv and std Rademacher complexity as an approximation of A and B, then A>B can be explained by the different constants of the two bounds, but not the weight norms (due to the same model).
>
> In summary, if we explain A>D by the Radamacher complexity bounds, then:
> (i) both the weight norm and the constants play a role;
> (ii) A>B can be explained by the different constants in our adversarial bound and the standard bound;
> (iii) B>D is explained by the different weight norms of the two models (standard-training v.s. adv-training), based on the standard bound.
>
> We can use simple formulas to explain.
>  Robust gen-gap $\approx$  constant_adv * weight_norm.
>  Standard gen-gap $\approx$  constant_std * weight _norm.
>  To explain A > B > D, by the above two bounds, we only need to explain:
>  constant_adv * weight_norm_adv > constant_std * weight_norm_adv > constant_std * weight_norm_std.
>  The first is due to constant_adv > constant_std, and the second is due to weight_norm_adv > weight_norm_std.
>
> We have added the related discussion in section 6 in the revised version. Thanks again for raising the question to help us figure out how to better explain the issue.
>
> Q2: Can we use some techniques to regularize this product during training to improve the generalization ability. The authors are recommended to give such kinds of experiments to support their theoretical results.
>
> A: Thanks for the suggestion. The explicit regularization of weight norms is essentially weight decay. We provide a new set of experiments of adversarial training with and without weight decay in Appendix C.4. We show that adversarial training without weight decay has a larger generalization gap and larger product of weight norms. This experiment also suggests the strong relationship between weight norms and the robust generalization gap.

---

### Official Review · Reviewer_Ma5T · 2021-11-04

**Correctness:** 4
**Technical Novelty And Significance:** 3
**Empirical Novelty And Significance:** 2
**Recommendation:** 8
**Confidence:** 4

**Main Review:**

The question investigated in the paper (described above) is natural and was posed as an open question by previous papers.

The main technique seems reasonable.

The experiments suggest that the change in the margin and the product of the weight norm may explain the larger generalization gap  (compared to the standard generalization).

Weakness: showing a depth/width-dependent lower bound could have been nice.

-Citing the paper https://arxiv.org/pdf/1810.02180.pdf (ALT19 and JMLR) is very relevant in the section "Adversarial Generalization".
It provides uniform convergence results In the case of a finite number of perturbations for a mixture of classifiers. The analysis goes through the Rademacher complexity as well.

-In section 3: d^2 should be 2^d?

**Summary Of The Paper:**

The main result of the paper is an upper bound on the Rademacher Complexity of neural networks in the case of adversarial examples (meaning, analyzing it with respect to the robust loss class).

Previous bounds for linear classifiers and shallow neural networks were investigated by Yin et al. (2019) and Awasthi et al. (2020).
The known methods for upper bounding the Rademacher Complexity for deep neural networks in the standard case don't apply in this case.

The main idea in this paper is to analyze the covering numbers directly (as opposed to calculating them by induction on the layers).


**Summary Of The Review:**

I recommend accepting the paper.

I read the overview of the main proof, but not the fully detailed proof.

---

> ### Author Response · Authors · 2021-11-17
> **We thank the reviewer for the comments and suggestions.**
>
> -------updated 11/19 11pm EST (mainly update answer to Q1)--------
>
> We thank the reviewer for the encouraging comments and suggestions. We appreciate the time you spent on the paper. Below we address the concerns and comments that you have provided.
>
> Q1: "Showing a depth/width-dependent lower bound could have been nice".
>
> A: Thanks for the suggestion. Bridging the gap between the upper and lower bound would be nice. There are two ways: one way is to show a depth/width-dependent lower bound as the reviewer suggested (increase lower bound); another way is to show a depth/width-independent upper bound (reduce upper bound). We briefly discuss which way is possible, and then discuss the technical challenges in both ways (though the reviewer only asked about the first way).
>
> 0) Which way is more likely to be true?
> If the upper bound can be improved to be depth-width-independent (thus matching our lower bound), then fundamentally the lower bound cannot be improved. Such a possibility exists. Actually, we are more inclined to this possibility, i.e., we tend to believe it is more promising to reduce the upper bound to be depth-width-independent, rather than increasing the lower bound. Anyhow, we don't have strong evidence of this possibility.
>
> 1) Technical challenge on increasing the lower bound (obtain a depth/width-dependent lower bound).
> In the current analysis, we construct a class of scalar networks to provide the lower bound. We obtain a closed-form expression of the adversarial examples. To obtain a depth/width-dependent lower bound, we need to: i) construct a more general function class of neural networks, and ii) then calculate the optimal adversarial examples in this class of neural networks. Currently, the challenge lies in the first step (construction). We have not tried hard to construct the function class so far, and we leave it to future work.
>
> 2) Can we reduce the upper bound (remove the dependence on depth/width in the upper bound)?
> This seems quite difficult by the current analysis. More specifically, the dependence h\sqrt{d log d} is probably unavoidable by our current approach of calculating the covering number.  Despite the technical difficulty, we suspect that reducing the upper bound is doable by using a new tool other than covering number and layer peeling. This is surely nontrivial and left as a future work. We add more discussions in Appendix D.
>
> Q2: Citing the paper https://arxiv.org/pdf/1810.02180.pdf (ALT19 and JMLR) is very relevant in the section "Adversarial Generalization". It provides uniform convergence results. In the case of a finite number of perturbations for a mixture of classifiers.
>
> A: Thanks a lot for the suggestion. This is indeed a relevant reference. We have now added this reference. In Section 3 and Appendix B in the revised paper, we added discussion on a few papers about VC dimension and uniform law of larger numbers, including the paper you mentioned.
>
> Q3: In section 3: d^2 should be 2^d?
>
> A: Thanks for pointing out the typo. We now fixed it in the revised paper.

---

### Author Response · Authors · 2021-11-17
**Summery of the revised version.**

We thank the reviewers for the comments and suggestions. We appreciate the time you spent on the paper. Below we summarize the change of our manuscript.

-------updated 11/22--------

section 1: We list our contributions in more detail.

section 2: We revised Proposition 1 and 2 in a more general form. We underline that convolution neural networks are also included in the hypothesis class.

section 3: The detailed discussion of the current methods is moved to Appendix B.1. We discuss related work, including the paper mentioned by the reviewers in the revised version.

section 4 to section 5.2: We fix some typos.

section 5.3: In the revised version, we group the factors into algorithm-independent factors (C_std, C_adv) and algorithm-dependent factors (W_std, W_adv).

section 6: In the revised version, we add the discussion about the robust generalization gap of a standard-trained model and the standard generalization gap of an adversarially-trained model. We further analyze why adversarial training cannot generalize well through the path C_adv W_adv > C_std W_adv > C_std W_std. For more detail, please see the paper.

Appendix A.3: We cite the results of the reference paper as proposition 4. We provide detailed proof of the lower bound.

Appendix B.1: We discuss the existing methods for Rademacher complexity in detail.

Appendix B.2: We carefully discuss the difficulty of applying layer peeling in adversarial settings.

Appendix B.3: We discuss why using covering numbers can overcome this issue.

Appendix B.4: We discuss adversarial generalization bounds in different frameworks.

Appendix C.4, We provide the experiments of weight decay.

Appendix D: We list the open problems.

---

### Decision · Program_Chairs · 2022-01-20

**Decision:**

Reject

**Comment:**

The paper made a solid theoretical contribution on the adversarial  generalization bounds of multi-layer neural networks.

However, the paper, at the current form, has many issues in the claim that "the product of the norm can explain the generalization gap":

(1). Weight decay. The authors uses the weight norm as the proxy for generalization gap, however, it is unclear to me that "adversarial trained networks have a larger generalization gap" can be explained by the product of weight norms. To carefully verify this, the authors have to at least carefully tune the weight decay, to the largest possible extend so the generalization error is not hurt, and compare the product of the weight norms in this scenario.  Without weight decay, the neural networks might learn a lot of redundancies in the weights (especially with adversarial training)  which makes the product of the norm to be too large.

The authors do perform experiments showing that with weight decay, the generalization gap becomes smaller and the norms become smaller, however, it is totally unclear to me that the weight decay considered in the experiments are actually optimal -- It could still be the case that with proper weight decay, the product of the norms in adversarial training is actually smaller comparing to that of the clean training.


Moreover, the authors should also clarify that **the product of the norms, according to the experiments, are simply too large and they can not be used in the theoretical result to get any meaningful generalization bounds**.



(2). The product of the norm in Rademacher complexity  is tight: This  claim only holds for neural networks with 1 neurons per layer. Once there are more than one neurons, there can simply be one neuron that learns f(x) and the other learns -f(x) and they completely cancels each other. So the product of the norm is obviously NOT TIGHT for any neural network with MORE than ONE NEURON per layer. In fact, the gap can be INFINITELY large.


Unfortunately, I like the paper very much and I hope this paper could be published, however, the claims  "the product of the norm can explain the generalization gap" is simply too misleading and ill-supported. I encourage the authors to completely remove this claim and submit the paper to COLT.

---

> ### Public Comment · ~Jiancong_Xiao1 · 2022-03-01
> **Clarification**
>
> Since the AC only provided two new criticisms in the meta-review and we have no chance to reply, we think it is necessary to answer these questions here.
> ___
> **Q1**, at least provide 'optimal' weight decay tuning for weight norms.
>
> A: 1, This experiment is related but not directly related to the main results of our paper, because people care more about optimal weight decay for performance. We cannot cover (even though a large-scale empirical research paper cannot cover) any cases in experiments. We can provide experiments the AC and reviewers want to see. After we received the meta-review, we ran this experiment proposed by the AC. We slightly increase the weight decay until the training fails. When the training fails, the weight norms are still larger than that of std-trained models. The report of the experiments is provided in the e-mail to the PCs, we hope it is forwarded to the AC.
>
> 2, From the words 'at least', it seems like this question means that we need to provide large-scale experiments to justify one claim. It is unfair for a theoretical paper. Moreover, our experiments are relatively large-scale. We have trained and reported 70 models to justify one claim. They cover different depths, different datasets, different numbers of training samples, w/o weight decay. As a comparison, many empirical research papers used fewer experiments to justify a claim. AC may think 'providing an explanation for generalization' is a huge topic, requiring large-scale experiments. But as we mentioned before, we can remove any claims and/or provide experiments as many as requested.
> ___
> **Q2**, The AC claimed that the 'weight norm is only tight for one neuron networks'.
>
> A: The AC's claim is wrong. Weight norm is tight for any-neuron networks.
>
> 1, We have already provided the proof in the paper. The AC did not point out a bug.
>
> 2, The AC's counterexample is not true. Let $F_2(P)=$ {$W_2\rho(W_1x): ||W_2||\cdot||W_1||\leq P $ }. $F_2'(P)=F_2(P)\cap A$, where $A$ is a set of additional constraint. In the AC's example, $A$ requires that the output of the first layer are $f(x)$ and $-f(x)$. Then, we may check that $Rad(F_2'(P))$ is equal to $0$.
>
> Since $F_2'(P)$ is a subset of $F_2(P)$, $Rad(F_2'(P))\leq Rad(F_2(P))$. Then, $Rad(F_2'(P))=0$ have no contradiction to $Rad(F_2(P))\geq\Theta(P)$.

---

> ### Public Comment · ~Jiancong_Xiao1 · 2022-03-01
> **Responds to the Meta-review**
>
> We are very surprised to receive the decision of "reject" for our Paper with reviewer scores of 8886. More surprisingly, the meta-review only contains two new criticisms, and the AC doesn't give us a chance to reply.  The two new criticisms are not reasonable and not convincing. The first one seems to ask for large-scale experiments for a theoretical paper. The second one is totally **wrong**. The AC used a wrong counterexample to 'disprove' our Theorem 3. (See clarification)
> ___
> Firstly, the meta-review is problematic.
>
> 1, It seems like the main reason for rejection is that the AC was not comfortable with the claim 'xx provides an explanation for generalization of xx.' Because AC said 'completely remove this claim and resubmit to COLT.'
>
> If this is the true reason for rejection, why didn't the AC raise it in any of the **three discussion phases** and let us remove it? Why didn't the AC say 'completely remove this claim in the final version' in the meta-review? This claim is not the main result of our paper, and we can simply remove it at any time. As far as we know, ICLR was created to be a more discussion-encouraging conference (as the founders are disappointed by other conferences' lack of communication).
>
> If this is not the true reason but some unsaid reason, we think the AC is unfair and negatively biased to our work.
>
> 2, The two new criticisms are not reasonable and not convincing. If these two are indeed what the AC is concerned about, AC should communicate with us before as we mentioned above.
>
> For the first one, we have already provided relatively large-scale experiments for a theory paper, and we can add as many as requested. It is okay if the AC thinks it is not enough. If the AC tells us that he wants to see weight decay experiments in the discussion phases, we can provide these experiments.
>
> For the second one, the counterexample is wrong. We think the AC is irresponsible because he posted a counterexample that is incorrect. If the AC thinks he has a counterexample, it is better to check it with us.
>
> Overall, we think the AC is **either irresponsible or negatively biased** to our work. The AC did not communicate with us and hid everything until the final decision to reject our paper. Additionally, we doubt what PCs said, 'The AC in charge of your paper is an expert on the topic ... We followed the AC's recommendation based on their strong expertise on the topic' in their reply on e-mail. We don't know what 'topic' the PC mentioned is. But we think the AC at least isn't familiar with classical Rademacher complexity works because of the simple mistake he made in criticism 2.
> ___
> Secondly, the decision procedure is not reasonable.
>
> When the AC wanted to raise new criticisms in meta-review to overrule a paper with scores of 8886, I think the SAC/PCs should check the correctness. If the SAC/PCs are not familiar with the sub-area, it is better to check it with us, with at least one of the four reviewers, or with an additional expert. In this case, the SAC/PCs overtrust the AC ('on their strong expertise on the topic'), and it is unfair to us.